# Exploring Knowledge Purification in Multi-Teacher Knowledge Distillation for LLMs

**Ruihan Jin**[1]     **Pengpeng Shao**[1,*]     **Zhengqi Wen**[1,*]     **Jinyang Wu**[1,*]
**Mingkuan Feng**[1]     **Shuo Yang**[1]     **Chu Yuan Zhang**[1]     **Jianhua Tao**[1,2,*]

[1]Department of Automation, Tsinghua University
[2]Beijing National Research Center for Information Science and Technology

`jinrh24@mails.tsinghua.edu.cn ppshao@tsinghua.edu.cn zqwen@tsinghua.edu.cn`
`wu-jy23@mails.tsinghua.edu.cn jhtao@tsinghua.edu.cn`

## Abstract

Knowledge distillation has emerged as a pivotal technique for transferring knowledge from stronger large language models (LLMs) to smaller, more efficient models. However, traditional distillation approaches face challenges related to knowledge conflicts and high resource demands, particularly when leveraging multiple teacher models. In this paper, we introduce the concept of **Knowledge Purification**, which consolidates the rationales from multiple teacher LLMs into a single rationale, thereby mitigating conflicts and enhancing efficiency. To investigate the effectiveness of knowledge purification, we further propose five purification methods from various perspectives. Our experiments demonstrate that these methods not only improve the performance of the distilled model but also effectively alleviate knowledge conflicts. Moreover, router-based methods exhibit robust generalization capabilities, underscoring the potential of innovative purification techniques in optimizing multi-teacher distillation and facilitating the practical deployment of powerful yet lightweight models.

## 1 Introduction

The rapid advancement of LLM has revolutionized various domains, including question answering (Yue, 2025) and reasoning (Plaat et al., 2024). The scaling law (Kaplan et al., 2020) unveils the correlation between the model size and generation capability, yet the practical deployment of colossal LLMs is often constrained by computational cost and resource demands, emphasizing the need for building efficient and lightweight models that preserve their power.

As the extension of model compression (Buciluǎ et al., 2006), knowledge distillation (Hinton et al., 2015) has emerged as a prominent solution to this challenge, which enables student models to inherit the capability of larger teacher models. Knowledge distillation is widely applied across various fields of machine learning (Kim & Rush, 2016; Park et al., 2019; Tang et al., 2019). To enhance knowledge diversity and specialized domain competencies, transferring knowledge from a multi-teacher ensemble to the student model attracts significant academic interest. This focus leads to the development of multi-teacher knowledge distillation approaches, such as TinyLLM (Tian et al., 2025) and TwT (Xu et al., 2025).

However, existing multi-teacher knowledge distillation frameworks suffer from two significant drawbacks: (1) *Knowledge Conflict*: Conflicting rationales among teacher LLMs are inevitable due to hallucinations, inconsistent reasoning paths or difference in expertise domains, impeding the effectiveness of knowledge transfer to the student model. This problem may become more pronounced as the number of teacher models increases. (2) *High Resource Demands*: Incorporating knowledge from multiple teachers inherently escalates resource requirements, necessitating complex sampling procedures and intricate training pipelines, which subsequently raises computational cost.

To investigate the adaptability of existing multi-teacher knowledge distillation frameworks toward more teacher LLMs, we perform extended experiments with TinyLLM (Tian et al., 2025), incre-

---

*Corresponding authors.

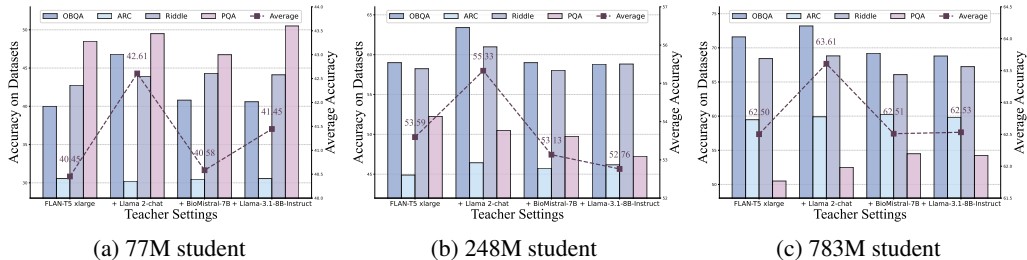

Figure 1: Effects of increasing teacher LLMs on the performance of the TinyLLM framework.

mentally introducing a series of teacher LLMs for distillation training(detailed in Appendix D.1). As illustrated in Fig. 1, contrary to our expectation that enlarging the teacher LLM ensemble would enhance the capabilities of student models, the distillation performance actually declines as the number of teacher models further increases. This decline indicates the detrimental impact of the knowledge conflict among teacher LLMs.

In this paper, we introduce the concept of **Knowledge Purification** in multi-teacher knowledge distillation. The core idea is to condense the knowledge of multiple teacher models from the rationale perspective. The knowledge purification integrates the rationales generated by multiple teacher LLMs into one single, consolidated rationale, which is subsequently employed during the distillation. Hence, the student model is provided with the rationale that encapsulates the collective insights of the teachers, enabling more efficient and effective distillation training. By purifying the knowledge, we mitigate the hallucinations and divergent reasoning paths among the teacher LLMs, thereby alleviating inter-teacher knowledge conflicts.

We further propose five methods to facilitate knowledge purification from distinct perspectives including aggregation, routing, and reinforcement learning (RL)-based selection. To thoroughly evaluate these approaches, we conduct extensive experiments on commonsense and biomedical reasoning tasks. Our results show that knowledge purification methods significantly enhance knowledge distillation performance across different student models and datasets. The effectiveness in alleviating knowledge conflicts is further verified. Furthermore, methods based on LLM routing demonstrate outstanding performance on out-of-domain datasets, underscoring the potential of utilizing knowledge purification to guide multi-teacher distillation across a broader spectrum.

Our contributions are summarized as follows:

- We identify the limitations of existing multi-teacher knowledge distillation frameworks, highlighting knowledge conflicts and high resource demands that hinder effective knowledge transfer.
- We introduce the concept of knowledge purification, which mitigates knowledge conflicts and enhances training efficiency by consolidating the rationales from multiple teachers into one coherent rationale. We propose five knowledge purification methods from different perspectives of aggregation, routing, and RL-based selection.
- Extensive experiments verify improvements of proposed methods in distillation performance and conflict mitigation. Further experiments on out-of-domain datasets illustrate the potential of knowledge purification in facilitating the generalization of multi-teacher knowledge distillation.

## 2 RELATED WORK

**Multi-Teacher Knowledge Distillation** Compared to utilizing single teacher, multi-teacher knowledge distillation harnesses broad knowledge diversity and rich reasoning paths, thereby enhancing the capabilities and generalization performance of student models (Liu et al., 2020; Zhang et al., 2024). TinyLLM (Tian et al., 2025) proposes a distillation paradigm that facilitates small student LLM to learn from rationales generated by two teacher LLMs. (Xu et al., 2025) incorporates rejection sampling and habitual reasoning in distillation to effectively balance computational cost and performance. These methods are constrained by knowledge conflicts among teacher LLMs, underscoring effective strategies for resolving these conflicts during distillation.

**LLM Routing**    In alignment with Mixture-of-Expert (MoE) (Jacobs et al., 1991; Collobert et al., 2003; Jiang et al., 2024), LLM routing aims to select the optimal LLM from diver candidates for a given question, enabling efficient activation of LLM ensembles. HybridLLM (Ding et al., 2024) leverages a hybrid approach to optimize both cost and quality for LLM pairs. Similarly, Router-LLM (Ong et al., 2024) explores effective methods for dynamic routing between a strong and a weak LLM. RouterDC (Chen et al., 2024) introduces dual contrastive learning and improves the routing performance. Recent innovations further explore the structured router (Jin et al., 2025) and the employment of reinforcement learning(Yue et al., 2025). Within the multi-teacher knowledge distillation framework, LLM routing presents a promising approach for effective knowledge purification.

## 3 FORMULATION

### 3.1 PRELIMINARIES

Generally, knowledge distillation uses the soft outputs/labels generated by the strong teacher model $T$ to transfer knowledge to a weak student model $S$. In this paper, we focus primarily on multiple choice question answering problems in NLP, utilizing LLMs as the main subjects of our study.

**Multiple Choice Question Answering**    In the $k$-multiple choice question answering task, given a question $q \in \mathcal{Q}$ and a corresponding candidate options set $\mathcal{O} = \{o_1, o_2, \ldots, o_k\}$, the objective of LLMs is to select the correct option from $\mathcal{O}$ that aligns with the ground truth option $o^* \in \mathcal{O}$. Besides, LLMs are encouraged to generate rationales, which have been shown to significantly enhance their performance (Wei et al., 2022). The answering process of an LLM $M$ (either the teacher $T$ or the student $S$) is formulated as:

$$o = M(q, \mathcal{O}, p_o), r = M(q, \mathcal{O}, p_r), \tag{1}$$

where $p_o$ and $p_r$ denote the prompt for predicting options and generating rationales, respectively.

**Multi-Teacher Knowledge Distillation**    We consider the rationale generated by the teacher LLM to be an embodiment of knowledge. We sample this rationale as $r_T = T(q, \mathcal{O}, p_r)$ from the teacher $T$ and construct the training set $\mathcal{D} = \{(q, \mathcal{O}, o^*, r_T)\}$ with $|\mathcal{D}|$ samples. The knowledge distillation for LLM leveraging rationales (Hsieh et al., 2023) can be formulated as:

$$\mathcal{L}_{\text{KD}} = \mathcal{L}_{\text{PR}} + \lambda \mathcal{L}_{\text{DL}}, \tag{2}$$

where $\lambda$ is a hyper-parameter balance between the prediction loss $\mathcal{L}_{\text{PR}}$ and the distillation loss $\mathcal{L}_{\text{DL}}$. The prediction loss $\mathcal{L}_{\text{PR}}$ guides the student to learn directly from ground truth options and the distillation loss $\mathcal{L}_{\text{DL}}$ supervises the student to inherit knowledge from the teacher's rationale. Details of the knowledge distillation for LLM are introduced in Appendix A.

Compared to the single-teacher approach, multi-teacher knowledge distillation leverages an ensemble of $n$ teacher LLMs $\mathcal{T} = \{T_1, T_2, \ldots, T_n\}$ to equip the student model with a broader spectrum of knowledge, leading to stronger generalization capabilities. In this context, we expand the training set to $\mathcal{D} = \{(q, \mathcal{O}, o^*, \mathcal{R})\}$, where $\mathcal{R} = \{r_{T_1}, r_{T_2}, \ldots, r_{T_n}\}$ denotes the rationales generated by each teacher LLM for the question $q$ and corresponding answer options $\mathcal{O}$. (Tian et al., 2025) extends the knowledge distillation framework to incorporate multiple teachers as:

$$\mathcal{L}_{\text{MTKD}} = \mathcal{L}_{\text{PR}} + \sum_{j=1}^{n} \lambda_j \mathcal{L}_{\text{DL}j}, \tag{3}$$

where $\mathcal{L}_{\text{DL}j}$ is the distillation loss with respect to the $j$-th teacher's rationale and $\lambda_j$ denotes the importance weight for $T_j$.

### 3.2 MOTIVATION ANALYSIS

Although most frameworks are originally designed to incorporate a fixed number of teacher LLMs for knowledge distillation, practically, we expect to enhance the capability and expertise of the student model by enlarging the teacher ensemble. We consider TinyLLM as a representative method

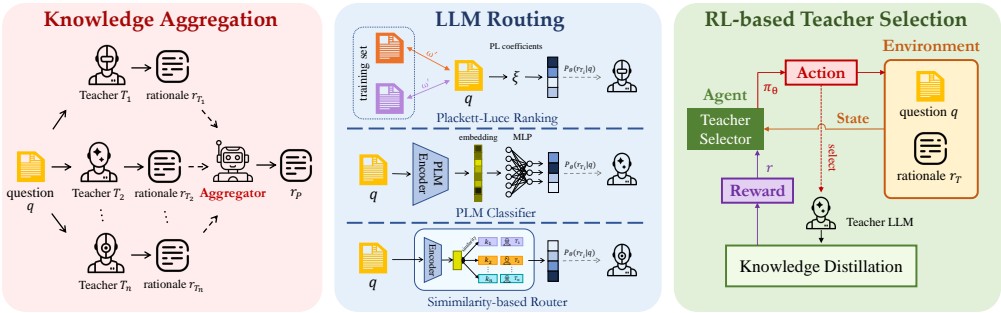

Figure 2: An illustration of five knowledge purification methods proposed in our work.

and conduct experiments to explore its adaptability to more teacher LLMs, as detailed in Appendix D.1. Results in Fig. 1 exhibit that, in these scenarios, the performance of TinyLLM significantly declines as the number of teacher LLMs further increases, which indicates the detrimental impact of knowledge conflicts among teacher LLMs. Furthermore, increasing the number of teachers can impose additional challenges related to computational resources and hyperparameter tuning. Therefore, there is an urgent need to develop a new framework to address these issues.

### 3.3 KNOWLEDGE PURIFICATION

In this section, we introduce the concept of **Knowledge Purification** in multi-teacher knowledge distillation. The knowledge purification process integrates the rationales generated by multiple teacher LLMs into one single, consolidated rationale, which is subsequently employed during the distillation. Specifically, given the rationales generated by each teacher LLM as $\mathcal{R} = \{r_{T_1}, r_{T_2}, \ldots, r_{T_n}\}$, the process of knowledge purification can be expressed as:

$$r_{\text{P}} = f(\mathcal{R}) = f(r_{T_1}, r_{T_2}, \ldots, r_{T_n}),\qquad(4)$$

where $f(\cdot)$ denotes the purification process and $r_{\text{P}}$ denotes the consolidated rationale. Through knowledge purification, we aim to mitigate knowledge conflicts among the teacher LLMs and enhance distillation efficiency. Our study investigates the impact of different purification methods on the performance of knowledge distillation.

Incorporating knowledge purification within the multi-teacher knowledge distillation framework alters the training objective, formulated as:

$$\mathcal{L}_{\text{MTKD-KP}} = \mathcal{L}_{\text{PR}} + \lambda \mathcal{L}_{\text{DL-KP}},\qquad(5)$$

where $\mathcal{L}_{\text{DL-KP}}$ denotes the distillation loss calculated using the consolidated rationale $r_{\text{P}}$:

$$\mathcal{L}_{\text{DL-KP}} = -\frac{1}{|\mathcal{D}|} \sum_{(q,\mathcal{O},\mathcal{R}) \in \mathcal{D}} \sum_{i=1}^{|r_{\text{P}}|} \log p(r_{\text{P}i}|r_{<i}, q, \mathcal{O}, p_r).\qquad(6)$$

## 4 METHODOLOGY

As shown in Fig. 2, we propose five methods to perform knowledge purification defined by Eq. 4.

**Knowledge Aggregation** We first consider performing knowledge purification by employing an aggregator, which is a global LLM that accepts all the rationales generated by teacher LLMs and combines the instructions to generate a consolidated rationale. We use an instruction-tuning paradigm (Wei et al., 2021) to provide instruction prompts containing in-context example as input and perform aggregation in a generation fashion.

**LLM Routing** Build upon a pool of candidate LLMs, an LLM router is designed to allocate an input question to the most appropriate LLM. Unlike aggregation, the key aspect of routing is selecting one rationale based on the probabilities predicted by the router as:

$$r_{\text{P}} = \arg\max_{r_{T_i}} P_\theta(r_{T_i}|q).\qquad(7)$$

In this paper, we design three representative LLM routing methods for knowledge purification:

- **Plackett-Luce ranking** We use a Plackett-Luce (PL) model (Luce et al., 1959; Plackett, 1975) for ranking multiple teacher LLMs. In the Plackett-Luce model, the probability of selecting a candidate rationale is modeled in a softmax relationship:

$$P_\theta(r_{T_i}|q) = \frac{e^{\xi_i}}{\sum_{j=1}^n e^{\xi_j}}, \quad i = 1, \ldots n, \tag{8}$$

We learn the PL coefficients $\xi = \{\xi_i : i = 1, \ldots n\}$ by solving:

$$\arg\min_\xi \sum_{q', \mathbf{y}_{gt}} [\omega' \cdot \ell(\mathbf{y}_{gt}, \frac{e^\xi}{\sum_{j=1}^n e^{\xi_j}})], \tag{9}$$

where $\ell$ denotes the cross-entropy loss, and $\mathbf{y}_{gt}$ denotes the ground truth label for the optimal selection. Inspired by (Ong et al., 2024), we use the weight $\omega'$ to measure the similarity between the input question $q$ and a question $q'$ in the database as $\omega' = \gamma^{1 + \frac{\epsilon \cdot \epsilon'}{\|\epsilon\| \cdot \|\epsilon'\|}}$, where $\gamma$ is a hyper-parameter, and $\epsilon$ and $\epsilon'$ denote text embeddings for $q$ and $q'$, respectively.

- **PLM classifier** We adopt a pre-trained language model (PLM) to extract textual features for standard text classification. Specifically, we employ a PLM encode the input question $q$ into a semantic embedding $h_{CLS}$ which is derived from the final hidden state corresponding to the special classification token (CLS). Subsequently, we use a two-layer perceptron to predict the probabilities of routing to each rationales $r_{T_i} \in \mathcal{R}$ as:

$$P_\theta(r_{T_i}|q) = \frac{e^{W_{i2}(W_{i1}h_{CLS}+b_{i1})+b_{i2}}}{\sum_{j=1}^n e^{W_{j2}(W_{j1}h_{CLS}+b_{j1})+b_{j2}}}, \quad i = 1, \ldots n, \tag{10}$$

where $W_i$ and $b_i$ denote the parameters of the MLP corresponding to $T_i$.

- **Similarity-based router** We follow RouterDC (Chen et al., 2024) to perform similarity-based LLM routing. We construct $n$ trainable LLM embeddings $\{\mathbf{k}_i : i \in 1, \ldots, n\}$ and calculate the cosine similarities between the question embedding and LLM embeddings for routing:

$$P_\theta(r_{T_i}|q) = \frac{e^{\text{sim}\langle \mathcal{E}(q), \mathbf{k}_i \rangle}}{\sum_{j=1}^n e^{\text{sim}\langle \mathcal{E}(q), \mathbf{k}_j \rangle}}, \quad i = 1, \ldots n, \tag{11}$$

where $\mathcal{E}$ denotes a language encoder, and $\text{sim}\langle \cdot, \cdot \rangle$ denotes the cosine similarity. The router is trained with two contrastive losses.

Details of all LLM routing methods are introduced in Appendix B.2.

**RL-based Teacher Selection** Inspired by (Yuan et al., 2021), we adopt a reinforcement learning (RL) framework to dynamically select teacher LLMs for knowledge purification. We define the state $s_i$ to encapsulate characteristics of the question $q$ and the rationales of the $i$-th teacher LLMs $r_{T_i}$ as:

$$s_i = [\mathcal{E}(q), \mathcal{E}(r_{T_i}) \cdot \mathbb{I}(T_i(q, \mathcal{O}, p_o) = o^*)] \in \mathbb{R}^{2d}, \tag{12}$$

where $\mathcal{E}$ is a language encoder, and $\mathbb{I}(T_i(q, \mathcal{O}, p_o) = o^*)$ indicates whether the teacher $T_i$ answers correctly. We design the policy function $\pi_\theta$ as:

$$\pi_\theta(s_i, a_i) = a_i \sigma(\mathbf{W}_i s_i + \mathbf{b}_i) + (1 - a_i)(1 - \sigma(\mathbf{W}_i s_i + \mathbf{b}_i)), \tag{13}$$

where $\sigma$ denotes the sigmoid function, and the action $a_i \in \{0, 1\}$ indicates whether to select the teacher $T_i$. Consider the definition of knowledge purification, we adopt the teacher LLM that receives the highest prediction score $\sigma(\mathbf{W}_i s_i + \mathbf{b}_i)$ and use it to guide the distillation process.

The trainable parameter of the teacher selector $\theta = \{\mathbf{W} \in \mathbb{R}^{n \times 2d}, \mathbf{b} \in \mathbb{R}^{n \times 1}\}$ is optimized using the standard policy gradient method:

$$\theta \leftarrow \theta + \beta \sum_i r \sum_{q \in \mathcal{Q}} \nabla_\theta \pi_\theta(s_i, a_i), \tag{14}$$

where $\beta$ denotes the learning rate. The reward $r = -\mathcal{L}_{PR} - \mathcal{L}_{DL}$ is computed based on the performance of the student model. During training, we alternately perform the knowledge distillation and the RL training. Detailed training algorithm is represented in Appendix B.3.

Table 1: Overall performance of multi-teacher knowledge distillation on four datasets. The best results across different datasets are highlighted in **bold**, with the second-best results are underlined. **Average** denotes average accuracy.

| Setting | Method | OBQA | ARC | Riddle | PQA | Average |
|---|---|---|---|---|---|---|
| *Teacher* | FLAN-T5 xlarge | 69.20 | 68.24 | 53.73 | 71.50 | 65.67 |
| | Llama 2-chat | 54.60 | 43.35 | 43.73 | 54.50 | 49.05 |
| | BioMistral-7B | 51.80 | 51.59 | 23.14 | 73.25 | 49.95 |
| | Llama-3.1-8B-Instruct | 65.60 | 71.67 | 60.98 | 75.00 | 68.31 |
| *FLAN-T5 small (77M) as Student* | Inference | 16.60 | 19.31 | 13.33 | 28.00 | 19.31 |
| | Fine-tuning | 45.60 | 31.76 | 49.22 | 47.50 | 43.52 |
| | Distilling-Step-by-Step | 41.00 | 30.73 | 44.90 | 49.25 | 41.47 |
| | TinyLLM | 40.60 | 30.56 | 44.12 | **54.25** | 42.38 |
| | Knowledge Aggregation | 40.40 | 30.47 | 43.92 | 53.25 | 42.01 |
| | Plackett-Luce Ranking | 42.00 | 31.76 | 45.69 | 50.50 | 42.49 |
| | PLM Classifier | 44.20 | 30.64 | 49.22 | 53.75 | 44.45 |
| | Similarity-based Router | **48.60** | **32.19** | **49.61** | 52.25 | **45.66** |
| | Teacher Selection | 46.80 | 30.39 | 48.82 | 52.50 | 44.63 |
| *FLAN-T5 base (248M) as Student* | Inference | 31.00 | 23.00 | 30.78 | 46.00 | 32.70 |
| | Fine-tuning | 61.00 | 43.61 | 56.86 | 51.00 | 53.12 |
| | Distilling-Step-by-Step | 59.00 | 46.01 | 59.41 | 52.50 | 54.23 |
| | TinyLLM | 58.80 | 46.18 | 58.82 | 47.25 | 52.76 |
| | Knowledge Aggregation | 58.00 | 45.75 | 58.43 | 51.50 | 53.42 |
| | Plackett-Luce Ranking | 62.40 | 46.27 | 58.63 | 54.75 | 55.51 |
| | PLM Classifier | 63.60 | 46.35 | 59.22 | **55.00** | 56.04 |
| | Similarity-based Router | **65.60** | 46.35 | 60.78 | 53.50 | 56.56 |
| | Teacher Selection | 65.00 | **46.70** | **61.76** | 53.25 | **56.68** |
| *FLAN-T5 large (783M) as Student* | Inference | 50.40 | 51.07 | 39.80 | 45.50 | 46.69 |
| | Fine-tuning | 72.00 | 60.26 | 67.45 | 53.00 | 63.18 |
| | Distilling-Step-by-Step | 71.60 | 60.00 | 68.43 | 51.00 | 62.76 |
| | TinyLLM | 68.80 | 59.83 | 67.25 | 54.25 | 62.53 |
| | Knowledge Aggregation | 71.40 | 59.74 | 68.63 | 53.50 | 63.32 |
| | Plackett-Luce Ranking | 72.60 | 59.48 | 69.41 | 56.50 | 64.50 |
| | PLM Classifier | 74.20 | 59.66 | 68.24 | 62.50 | 66.40 |
| | Similarity-based Router | **76.60** | 60.60 | **70.59** | 61.00 | 67.20 |
| | Teacher Selection | 75.20 | **61.12** | 70.39 | **63.50** | **67.55** |

## 5 EXPERIMENTS

**Models** We consider a multi-teacher ensemble of four LLMs: FLAN-T5 xlarge (2.85B), Llama 2-chat (Touvron et al., 2023)(7B), BioMistral-7B (Labrak et al., 2024), and Llama-3.1-8B-Instruct (Dubey et al., 2024). We conduct experiments using FLAN-T5 (Chung et al., 2024) small (77M), base (248M), and large (783M) as student models, respectively.

**Datasets** We conduct experiments on four multiple choice question answering datasets in commonsense reasoning and biomedical reasoning. For **commonsense reasoning**, we consider Open-BookQA (OBQA) (Mihaylov et al., 2018), AI2 Reasoning Challenge (ARC) (Clark et al., 2018), and RiddleSense (Riddle) (Lin et al., 2021). For **biomedical reasoning**, we consider PubMedQA (PQA) (Jin et al., 2019). To ensure a fair comparison among knowledge purification methods, we randomly retain 80% of the training set samples for distillation training, and the remaining 20% serve as the public set. A joint dataset, composed of the public set of each dataset, is utilized for training LLM routers. Details of dataset construction are provided in Appendix C.2.

**Metrics** We calculate the accuracy of the distilled student model in multiple choice question answering tasks as the primary performance metric:

$$\text{ACC} = \frac{1}{|\mathcal{Q}|} \sum_{q \in \mathcal{Q}} \mathbb{I}(S(q, \mathcal{O}, p_o) = o^*). \tag{15}$$

Table 2: Comparison of knowledge purification methods from a practical perspective. Each method is analyzed in: **Prior**, **Parameters**, **Training Necessity**, **Transferability**, and **Latency**.

| Method | Prior | Parameters | Training Necessity | Transferability | Latency |
|---|---|---|---|---|---|
| Knowledge Aggregation | $q, \mathcal{R}$ | >10B | ✗ | ✓ | s |
| Plackett-Luce Ranking | $q$ | ~278M | ✗ | ✓ | s |
| PLM Classifier | $q$ | ~278M | ✓ | ✓ | ms |
| Similarity-based Router | $q$ | ~278M | ✓ | ✓ | ms |
| Teacher Selection | $q, \mathcal{R}$ | ~278M | ✓ | ✗ | min |

Furthermore, we assess the effectiveness of knowledge purification methods in mitigating knowledge conflicts among teacher LLMs by calculating the *Conflict Mitigation Value* (CMV). We define CMV as the average accuracy improvement achieved through knowledge purification in distillation training with a series of incremental teacher LLMs, compared to the baseline TinyLLM framework:

$$\text{CMV} = \frac{1}{n-1} \sum_{i=2}^{n} (\text{ACC}_{\text{KP}, |\mathcal{T}|=i} - \text{ACC}_{\text{TinyLLM}, |\mathcal{T}|=i}). \tag{16}$$

**Implementation details** For knowledge distillation, we use the AdamW (Loshchilov & Hutter, 2019) optimizer and set the learning rate to $5 \times 10^{-5}$, the batch size to 8, and the maximum input length to 512. We find that $\lambda = 4$ works best for balanced training. We employ GPT-4 (Achiam et al., 2023) as the knowledge aggregator. mDeBERTaV3-base (He et al., 2021) is adopted as the language encoder for the PLM classifier, the similarity-based router and the RL-based teacher selector. Details of applying knowledge purification approaches are included in Appendix B. All experiments are conducted on four NVIDIA A100 80GB GPUs.

## 5.1 RESULTS

We conduct comprehensive experiments to evaluate the performance of multi-teacher knowledge distillation employing knowledge purification methods. Our comparison includes the proposed methods against the original teacher LLMs and four baseline approaches. These baselines consist of direct inference, fully fine-tuning the student model, Distilling-Step-by-Step (Hsieh et al., 2023) which leverages teacher's rationale as additional supervision, and TinyLLM (Tian et al., 2025) which integrates all rationales generated by all teacher LLMs. Details of baseline approaches are included in Appendix C.3. The overall experimental results are presented in Tab. 1.

From a performance perspective, no single method demonstrates a significant advantage over the others. Overall, the similarity-based router and the RL-based teacher selection consistently achieve the highest average accuracies, ranking either first or second across the distillation experiments involving all three student models. For distilling the FLAN-T5 small model, the similarity-based router attains the highest average accuracy of 45.66%, exceeding baselines by at least 4.9%. For distilling the FLAN-T5 base and FLAN-T5 large models, the RL-based teacher selection exhibits superior performance, surpassing the best baseline by 4.5% and 6.9%, respectively.

Generally, the implementation of the knowledge purification method demonstrates advantages over the baseline, underscoring its positive impact on knowledge distillation. The performance of LLM routing is exemplary, while the PL ranking shows slightly weaker results compared to the other two LLM routers. In contrast, the overall performance of the knowledge aggregation method remains relatively inferior, with no significant improvement observed. This suggests that, despite the strong capacity of the aggregator (e.g., GPT-4), the enhancing effect of the consolidated rationale through aggregation on knowledge distillation remains uncertain.

Compared to the teacher LLMs, the distilled 783M student model exceeds the average accuracy of three teachers and ranks second only to Llama-3.1-8B-Instruct, illustrating the effectiveness of knowledge purification. Additionally, we observe that knowledge purification yields more substantial improvements in larger student models. This can be attributed to the stronger capacity of larger models to learn from the generated rationales, leading to the notable outcomes achieved through our targeted purification approaches. In contrast, smaller models tend to focus primarily on fitting the final option, which limits the enhancements gained from knowledge purification.

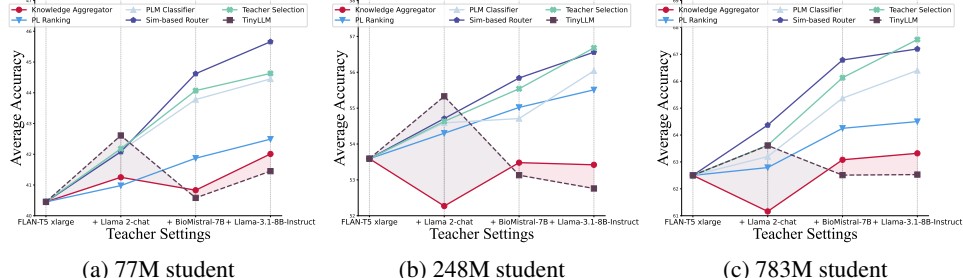

| (a) 77M student | (b) 248M student | (c) 783M student |

Figure 3: Evaluation of knowledge purification methods with an increasing number of teacher LLMs. We visualize the CMV of knowledge aggregation as an example, represented by the signed area of the shaded region (positive when above TinyLLM and negative when below).

Table 3: Adaptability of knowledge purification methods toward multiple teacher LLMs, evaluated in *Conflict Mitigation Value* (CMV).

| Method | CMV$_{77M\ student}$ | CMV$_{248M\ student}$ | CMV$_{783M\ student}$ |
|---|---|---|---|
| Knowledge Aggregation | −0.003 | −0.007 | −0.004 |
| Plackett-Luce Ranking | +0.001 | +0.012 | +0.010 |
| PLM Classifier | +0.018 | +0.014 | +0.021 |
| Similarity-based Router | **+0.025** | **+0.020** | **+0.032** |
| Teacher Selection | +0.020 | +0.019 | +0.029 |

## 5.2 QUALITATIVE ANALYSIS ON KNOWLEDGE PURIFICATION METHODS

In this section, we perform a systematic analysis on the proposed knowledge purification methods from a practical perspective. We consider the following metrics to evaluate each methods: **Prior** refers to the input of each model; **Parameters** refers to additional amount of parameters introduced for purification; **Training Necessity** refers to whether training is required; **Transferability** refers to whether the method can be applied to a new dataset without requiring additional training; and **Latency** refers to the time scale required for processing a single instance of knowledge purification.

We present our analysis in Tab. 2. For priors, LLM routing methods require only the question as input and do not necessitate pre-sampling rationales from the teacher LLM. This enables us to leverage pretrained LLM routers to guide rationale sampling in multi-teacher knowledge distillation as further demonstrated in Section 5.4. Aside from knowledge aggregation, which employs a strong LLM (e.g., GPT-4) for synthesis, other methods only introduce additional parameters on the size of the PLM. The training process of the teacher selector is closely coupled with rewards from knowledge distillation, necessitating retraining when applied to new datasets. This training also introduces a significant delay in the implementation of RL-based teacher selection, considerably exceeding the millisecond-level delay of the PLM classifier and the similarity-based router.

## 5.3 ADAPTABILITY TOWARD MULTIPLE TEACHER LLMS

We further evaluate the adaptability of knowledge purification methods in distillation training with a series of incremental teacher LLMs. We adopt the same experimental setup we used when revealing the knowledge conflict of TinyLLM. The evaluation result is visualized in Fig. 3 (detailed in Appendix D.2). Tab. 3 demonstrates the CMV of each method in mitigating knowledge conflicts.

We observe that applying knowledge aggregation reports negative CMV for all three student models, suggesting that it fails to effectively mitigate knowledge conflicts. In contrast, all LLM routing methods and RL-based teacher selection report positive CMV, indicating their potential in alleviating knowledge conflicts. Notably, the similarity-based router achieves the highest CMV across all three student models, demonstrating its superior adaptability to an incremental multi-teacher ensemble.

## 5.4 ROUTER-GUIDED OUT-OF-DOMAIN KNOWLEDGE DISTILLATION

Grounded in knowledge purification, we perform out-of-domain knowledge distillation. We exclude the knowledge aggregation which does not involve training and RL-based teacher selection with

Table 4: Experimental results on *out-of-domain* datasets. We verify the potential of utilizing LLM routing methods to guide multi-teacher knowledge distillation on out-of-domain data. The best results are highlighted in **bold**, with the second-best results are underlined.

| Setting | Method | PIQA | BioASQ |
|---|---|---|---|
| *Teacher* | FLAN-T5 xlarge | 58.43 | 65.85 |
| | Llama 2-chat | 60.50 | 69.92 |
| | BioMistral-7B | 67.46 | 90.24 |
| | Llama-3.1-8B-Instruct | 70.84 | 88.62 |
| FLAN-T5 small (77M) *as Student* | Inference | 20.78 | 47.97 |
| | Fine-tuning | 42.33 | 78.86 |
| | Distilling-Step-by-Step | 49.29 | 81.30 |
| | TinyLLM | 49.84 | 75.61 |
| | Plackett-Luce Ranking | 52.77 | 80.47 |
| | PLM Classifier | 50.05 | **82.11** |
| | Similarity-based Router | **53.97** | **82.11** |
| FLAN-T5 base (248M) *as Student* | Inference | 30.47 | 57.72 |
| | Fine-tuning | 47.55 | 89.43 |
| | Distilling-Step-by-Step | 55.93 | 86.18 |
| | TinyLLM | 56.80 | 78.05 |
| | Plackett-Luce Ranking | **63.98** | 86.99 |
| | PLM Classifier | 59.63 | 85.37 |
| | Similarity-based Router | 63.33 | **90.24** |
| FLAN-T5 large (783M) *as Student* | Inference | 51.90 | 63.41 |
| | Fine-tuning | 58.43 | 90.24 |
| | Distilling-Step-by-Step | 60.72 | 86.99 |
| | TinyLLM | 68.88 | 82.93 |
| | Plackett-Luce Ranking | 68.77 | 87.80 |
| | PLM Classifier | 67.90 | 83.74 |
| | Similarity-based Router | **69.53** | **91.87** |

limited transferability. Instead, we focus on the proposed LLM routing approaches, as the generalization ability for out-of-domain data is a crucial metric for assessing the effectiveness of LLM routers. Physical Interaction Question Answering (PIQA)(Bisk et al., 2020) and BioASQ (Tsatsaronis et al., 2015) serve as two out-of-domain datasets, which represent commonsense reasoning and biomedical reasoning, respectively. As illustrated in Tab. 4, utilizing LLM routers to guide knowledge distillation yields strong generalization ability. The similarity-based router achieves the highest accuracy across most settings, significantly exceeding baselines. Besides, the PL ranking outperforms the PLM classifier in overall performance and demonstrates robust generalization.

It is worthy noting that LLM routing approaches only require the question as input, eliminating the need for pre-sampling responses from teacher LLMs. When applied to a broader spectrum of out-of-domain data, high-performing LLM routers can effectively direct the sampling process of a multi-teacher ensemble, thereby facilitating subsequent knowledge distillation. This approach significantly reduces computational costs and resource consumption during the sampling phase while effectively alleviating knowledge conflicts and enhancing the performance of the distilled model. This presents a promising framework for the rapid and flexible implementation of multi-teacher knowledge distillation and the deployment of powerful yet lightweight models.

## 5.5 EFFICIENCY OF KNOWLEDGE DISTILLATION APPROACHES

In addition to performance evaluation, our experiments aim to evaluate the improvement offered by the knowledge purification methods concerning the efficiency of multi-teacher knowledge distillation. To ensure comparability, we fix the distillation epoch at 4000 and evaluate the training efficiency of different methods for distilling the FLAN-T5 large model on the ARC dataset, utilizing GPU hours as the quantitative metric. For knowledge purification methods, we consider the computational consumption of both the purification stage (e.g., aggregation, training the router) and the distillation stage. The results are exhibited in Tab. 5

Table 5: Efficiency of different methods for distilling the FLAN-T5 large model on the ARC dataset. We consider both purification and distillation stages and utilize GPU hours as the metric.

| Method | $|\mathcal{T}|$ | Purification Stage | Distillation Stage | Total |
|---|---|---|---|---|
| Fine-tuning | 1 | - | 0.7 | 0.7 |
| Distilling-Step-by-Step | 1 | - | 1.1 | 1.1 |
| TinyLLM | 4 | - | 2.6 | 2.6 |
| Knowledge Aggregation | 4 | 5.2[1] | 1.5 | 6.7 |
| Plackett-Luce Ranking | 4 | 0.1 | 1.3 | 1.4 |
| PLMClassifier | 4 | 0.5 | 1.2 | 1.7 |
| Similarity-based Router | 4 | 0.6 | 1.2 | 1.8 |
| Teacher Selection | 4 | 3.5 | | 3.5 |

Among the proposed knowledge purification methods, knowledge aggregation demonstrates the highest GPU comsumption due to the extensive call of the open-source LLM as the aggregator. When a proprietary LLM, such as GPT-4, is employed as the aggregator, its inefficiency is reflected in the equally significant latency and elevated costs. The RL-based teacher selection uniformly optimizes the purification and distillation, while its iterative training entails considerable computational consumption. Conversely, routing-based methods significantly enhance the efficiency of knowledge distillation compared to TinyLLM, which employs all rationales. Notably, training the LLM router demands fewer computational resources than the distillation process. Furthermore, given the generalization capabilities of routers, the impact of knowledge purification on efficiency enhancement is expected to be more pronounced when applied to out-of-domain data.

## 6 LIMITATIONS

Due to the limited computational resources, we only construct a teacher ensemble of four LLMs. While we strategically select teacher LLMs — such as BioMistral-7B — to supply domain-specific knowledge in biomedical reasoning, it remains challenging to guarantee that the knowledge represented by the teacher ensemble is comprehensive. In the practical application of multi-teacher knowledge distillation, a larger number of teacher models would be more conducive to enhancing the specialized domain capabilities of student models. The restricted number of teacher LLMs currently limits our ability to thoroughly assess the effectiveness of knowledge purification methods. Although we conduct a small-scale evaluation involving six teachers, as presented in Appendix D.3, further evaluations will be necessary to fully explore and validate the adaptability.

Knowledge distillation is a universal framework for transferring knowledge from powerful models to weak ones. In this paper, we primarily focus on the NLP field and consider LLMs as the main subjects of our study. The proposed knowledge purification methods are tailored to the characteristics of LLM. Approaches such as LLM routing and teacher selection have the potential to generalize to broader machine learning tasks, but specific implementation and evaluation still require further investigation. We leave the investigation of such scenarios to future work.

## 7 CONCLUSION

In this paper, we tackle the challenges inherent in multi-teacher knowledge distillation frameworks, specifically addressing knowledge conflicts and high resource demands that hinder effective knowledge transfer to student models. We introduce the concept of **Knowledge Purification**, aimed at reducing divergent reasoning paths among teachers and enhancing distillation efficiency by consolidating the rationales from multiple teacher LLMs. We propose five methods to facilitate knowledge purification from distinct perspectives. Extensive experiments across commonsense and biomedical reasoning tasks demonstrate that proposed methods significantly enhance the performance of distilled models while effectively mitigating knowledge conflicts. Notably, the approach based on LLM routers showed exceptional performance on out-of-domain datasets, underscoring its broad applicability and practical value. In summary, our findings contribute to the advancement of multi-teacher knowledge distillation frameworks, paving the way for the practical deployment of efficient and powerful lightweight models.

---

[1]The GPU hour is accessed when using Llama-3.1-70b as the aggregator.

ACKNOWLEDGMENTS

This work is supported by the National Natural Science Foundation of China under Grants U2436210.

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

# A DETAILS OF KNOWLEDGE DISTILLATION FOR LLM

Knowledge distillation (Hinton et al., 2015) is designed to facilitate the transfer of knowledge from the teacher model to the student model. Unlike traditional deep learning, where soft labels can be obtained from teacher models, LLMs are often treated as black-box models. In this context, we typically regard the rationales generated by the teacher LLM as the embodiment of knowledge.(Hsieh et al., 2023) We sample the rationale generated by the teacher LLM $T$ as $r_T = T(q, \mathcal{O}, p_r)$ and construct the training set $\mathcal{D} = \{(q, \mathcal{O}, o^*, r_T)\}$ with $|\mathcal{D}|$ samples. We expect the student model to inherit knowledge from the teacher's rationale and supervise this process using distillation loss, which is defined in the form of the cross-entropy loss as:

$$\mathcal{L}_{\text{DL}} = -\frac{1}{|\mathcal{D}|} \sum_{(q, \mathcal{O}, r_T) \in \mathcal{D}} \sum_{i=1}^{|r_T|} \log p(r_{Ti} | r_{<i}, q, \mathcal{O}, p_r), \quad (17)$$

where $|r_T|$ denotes the number of tokens in the teacher's rationale, and $p(r_{Ti} | r_{<i}, q, \mathcal{O}, p_r)$ denotes the probability of generating token $r_{Ti}$, given the inputs and the already generated tokens. Besides, a prediction loss is introduced in training the student model as:

$$\mathcal{L}_{\text{PR}} = -\frac{1}{|\mathcal{D}|} \sum_{(q, \mathcal{O}, o^*) \in \mathcal{D}} \sum_{i=1}^{|o^*|} \log p(o_i^* | o_{<i}, q, \mathcal{O}, p_r), \quad (18)$$

where $|o^*|$ denotes the number of tokens in the ground truth option, and $p(r_{Ti} | r_{<i}, q, \mathcal{O}, p_r)$ denotes the probability of generating token $o_i^*$, given the input and the already generated tokens.

The global knowledge distillation process is formulated as:

$$\mathcal{L}_{\text{KD}} = \mathcal{L}_{\text{PR}} + \lambda \mathcal{L}_{\text{DL}}, \quad (19)$$

where $\lambda$ is a hyper-parameter balance between the prediction loss $\mathcal{L}_{\text{PR}}$ and the distillation loss $\mathcal{L}_{\text{DL}}$.

# B DETAILS OF KNOWLEDGE PURIFICATION APPROACHES

## B.1 KNOWLEDGE AGGREGATION

We employ the powerful proprietary LLM — GPT-4 (Achiam et al., 2023) as the knowledge aggregator and use an instruction-tuning paradigm (Wei et al., 2021) to guide GPT-4 to perform aggregation in a generation fashion. Fig. 4 shows the prompt used for knowledge aggregation. In practice, we pre-label 10 knowledge aggregation samples and randomly select one as the in-context example during inference.

In addition to the default GPT-4, we also consider the comparatively weaker Llama-3.1-70b (Dubey et al., 2024) as the aggregator to compare the impact of different aggregator choices. For both aggregators, we adopt the same prompt format and examine the performance of applying knowledge aggregation on distilling the FLAN-T5 large student model. The results, presented in Tab. 6, indicate that utilizing a more powerful model as the aggregator does not lead to a significant enhancement in performance.

## B.2 LLM ROUTING

**Plackett-Luce ranking** The Plackett-Luce (PL) model was initially introduced by Plackett (Plackett, 1975) to rank the horses in gambling. It has since been applied to describe the processes of ranking and selecting multiple candidate items in various domains. Notably, when the number of items is limited to two, the Plackett-Luce model simplifies to the Bradley-Terry (BT) model (Bradley & Terry, 1952). Chatbot Arena Platform (Chiang et al., 2024) is built based on the ranking of LLMs by the Bradley-Terry model. The ranking is formulated as estimating the Bradley-Terry coefficient $\xi$:

$$\arg\min_{\xi} \mathbb{E}_{(A,H)}[\ell(H, \frac{1}{1 + e^{\xi_{A_1} - \xi_{A_2}}})], \quad (20)$$

```
# Instruction
As an expert synthesizer of information, you possess the ability to distill complex ideas into clear and concise
insights. Your task is to analyze and summarize a series of reasoning rationales into a single cohesive rationale.
Please ensure that the final summary captures the essential elements and nuances of the original reasoning, while
maintaining clarity and coherence.
# In-Context Example
Here is an example:
Input: Question: An astronomer observes that a planet rotates faster after a meteorite impact. Which is the most
likely effect of this increase in rotation?\nAnswer Choices:\n(a) Planetary density will decrease.\n(b) Planetary
years will become longer.\n(c) Planetary days will become shorter.\n(d) Planetary gravity will become stronger.\n
1. (c) Planetary days will become shorter. After a meteorite impact, the rotation of the Earth is increased.
Planetary days will become shorter after a meteorite impact.
2. Let's think step by step.\nWhat happens to the planetary rotation after the impact?\nRotation of the planet
will increase.\nWhat happens to the planetary days after the impact?\nDays will become shorter.\nWhat happens to
the planetary years after the impact?\nPlanetary years will become longer.\nWhat happens to the planetary gravity
after the impact?\nGravity will become stronger.\nOption (a) is incorrect.\nOption (b) is incorrect.\nOption (d)
3. To solve this question, let's break it down step by step:\n\n1. **Initial Condition**: A planet has a certain
angular momentum due to its rotation.\n2. **Meteorite Impact**: When a meteorite impacts the planet, it transfers
some of its momentum to the planet.\n3. **Transfer of Momentum**: This transferred momentum is in the form of
linear momentum, which also includes an angular component due to the impact's location and angle.\n4.
**Conservation of Angular Momentum**: The total angular momentum of a closed system remains constant. In this
case, the planet's angular momentum increases due to the transferred momentum.\n5. **Effect of Increased Angular
Momentum**: An increase in angular momentum results in a faster rotation period, as the planet's rotation speed
increases.\n6.     **Result**:     If     the     planet's     rotation     period     decreases,     its     days     will     become
shorter.\n\nConsidering the answer choices:\n\n- (a) Planetary density will decrease: This is unlikely, as the
impact does not necessarily affect the planet's overall density.\n- (b) Planetary years will become longer: This
is incorrect, as a shorter rotation period would result in shorter years, not longer.\n- (c) Planetary days will
become shorter: This is the correct answer, as the increased rotation
4. The most likely effect of this increase in rotation is that (c) planetary days will become shorter. The
reasoning behind this is that the impact of the meteorite would transfer energy to the planet, causing it to
rotate faster. This increase in rotation would result in shorter planetary days.
Create a cohesive rationale that captures the essential points from each model while ensuring clarity and logical
flow. Output: The observed increase in a planet's rotation following a meteorite impact leads to the conclusion
that planetary days will become shorter. This occurs due to the transfer of momentum from the meteorite to the
planet, which enhances its angular momentum. According to the principle of conservation of angular momentum, an
increase in rotation speed results in a shorter rotation period.\nTo clarify the implications of this
phenomenon:\n1. **Initial State**: The planet has a defined angular momentum based on its current rotation.\n2.
**Impact Event**: The meteorite impacts the planet, imparting additional momentum.\n3. **Angular Momentum
Conservation**: The total angular momentum remains constant, but the planet's rotation speed increases due to the
added momentum.\n4. **Resulting Effect**: With an increased rotation speed, the duration of a planetary day is
reduced.\n\nTherefore, the correct answer is that planetary days will become shorter (option c). Other options,
such as changes in density, lengthening of years, or strengthening of gravity, do not accurately reflect the
consequences of the increased rotation from the impact.
# Query
Input: Question: {QUESTION}
1. {T5_RATIONALE}
2. {LLAMA2_RATIONALE}
3. {LLAMA3_RATIONALE}
4. {BIOMISTRAL_RATIONALE}
Create a cohesive rationale that captures the essential points from each model while ensuring clarity and logical
flow. Output:
```

Figure 4: Prompt used for GPT-4 to perform knowledge aggregation, consisted of global instruction, in-context example, and query.

where $\ell$ denotes the binary cross-entropy loss. $A$ and $H$ denote the LLM pair and the human response. In our work, we consider the Plackett-Luce ranking and extend the framework to multi-teacher routing. We transform the problem into a cross-entropy loss optimization problem weighted by text similarity. We learn the PL coefficients $\xi = \{\xi_i : i = 1, \dots n\}$ by solving Eq. 9, where $\mathbf{y}_{\text{gt}}$ denotes the ground truth label for the optimal selection of the rationales and is represented in an one-hot encoding format. We consider the description generated by the teacher model that produces the minimum number of tokens while making correct selections as the optimal choice, taking into account the preference for computational efficiency and cost reduction in practical applications. $\omega' = \gamma^{1+\frac{\epsilon \cdot \epsilon'}{\|\epsilon\| \cdot \|\epsilon'\|}}$ measures the similarity between the input question $q$ and a question $q'$ in the database. Following (Ong et al., 2024), we adopt the exponential scale and choose $\gamma = 10$.

It is important to note that no training is necessary for the ranking, and all computations are performed during inference. We will elaborate on the specific inference.

**Proposition 1.** *Let $\xi$ satisfy Eq. 9. Then, it satisfies the following condition:*

$$\frac{e^{\xi_i}}{\sum_{j=1}^n e^{\xi_j}} = \frac{\sum_{q' \in \mathcal{Q}_i} \omega'}{\sum_{q'} \omega'}, \quad i = 1, \dots n,$$

$$\text{s.t.} \quad \mathcal{Q}_i = \{q' : T_i \text{ is optimal for } q'.\} \tag{21}$$

*Proof.* Note:

$$\mathbf{y}(i) = \frac{e^{\xi_i}}{\sum_{j=1}^n e^{\xi_j}}, \quad i = 1, \dots n,$$

Table 6: Comparison of different aggregators for **Knowledge Aggregation**. The FLAN-T5 large model serves as the student model.

| Aggregator | OBQA | ARC | Riddle | PQA | Average |
|---|---|---|---|---|---|
| Llama-3.1-70b | 71.20 | 60.77 | 67.65 | 52.50 | 63.03 |
| GPT-4 | 71.40 | 59.74 | 68.63 | 53.50 | 63.32 |

$$g(\mathbf{y}) = \sum_{q',\mathbf{y}_{\text{gt}}} [\omega' \cdot \ell(\mathbf{y}_{\text{gt}}, \frac{e^{\xi}}{\sum_{j=1}^{n} e^{\xi_j}})] = \sum_{q',\mathbf{y}_{\text{gt}}} [\omega' \cdot \ell(\mathbf{y}_{\text{gt}}, \mathbf{y})],$$

$$h(\mathbf{y}) = (\sum_{i=1}^{n} \mathbf{y}(i)) - 1.$$

To maximize $g(\mathbf{y})$ subject to $h(\mathbf{y}) = 0$, we consider the Lagrangian:

$$\mathcal{L}(\mathbf{y}, \eta) = g(\mathbf{y}) + \eta h(\mathbf{y}),$$

where $\eta$ is the Lagrange multiplier. We take the partial derivatives of $\mathcal{L}$ with respect to $\mathbf{y}$ and $\eta$, and set them to zero:

$$\frac{\partial \mathcal{L}}{\partial \mathbf{y}(i)} = 0, \frac{\partial \mathcal{L}}{\partial \eta} = 0.$$

This yields the system of equations:

$$\sum_{q' \in \mathcal{Q}_i} \omega' \frac{\partial \log \mathbf{y}(i)}{\partial \mathbf{y}(i)} - \eta = 0, \quad i = 1, \ldots n,$$

$$(\sum_{i=1}^{n} \mathbf{y}(i)) - 1 = 0.$$

From this, we derive:

$$\frac{e^{\xi_i}}{\sum_{j=1}^{n} e^{\xi_j}} = \mathbf{y}(i) = \frac{\sum_{q' \in \mathcal{Q}_i} \omega'}{\sum_i \sum_{q' \in \mathcal{Q}_i} \omega'} = \frac{\sum_{q' \in \mathcal{Q}_i} \omega'}{\sum_{q'} \omega'}, \quad i = 1, \ldots n.$$

Thus, it satisfies the condition in Eq. 21. □

In practical inference, we perform Plackett-Luce ranking based on Eq. 21.

**PLM classifier** For training the PLM classifier, the definition of the ground truth label $\mathbf{y}_{\text{gt}}$ we use is consistent with that in the Plackett-Luce ranking. We utilize mDeBERTaV3-base (He et al., 2021) as the language encoder and extract the semantic embedding $h_{\text{CLS}}$ for the input question. We use a two-layer MLP with a hidden layer dimension of 128 to predict the routing probabilities based on $h_{\text{CLS}}$. We perform full-parameter training and train the classifier for 5000 epochs, using the AdamW (Loshchilov & Hutter, 2019) optimizer with batch size 16 and learning rate $5 \times 10^{-5}$.

**Similarity-based router** We follow RouterDC (Chen et al., 2024) to perform similarity-based LLM routing and adopt two contrastive losses to train the router. For each question in training, we assign a binary score in $\{0, 1\}$ to the LLM based on correctness, and sample the LLM with the highest and lowest scores respectively (randomly selecting one in the case of ties) to calculate the sample-LLM contrastive loss as:

$$\mathcal{L}_{\text{sample-LLM}} = \sum_q -\log \frac{e^{\text{sim}\langle \mathcal{E}(q), \mathbf{k}^+ \rangle}}{e^{\text{sim}\langle \mathcal{E}(q), \mathbf{k}^+ \rangle} + e^{\text{sim}\langle \mathcal{E}(q), \mathbf{k}^- \rangle}}, \quad (22)$$

where $\mathbf{k}^+$ and $\mathbf{k}^-$ denote the LLM embedding of the LLM with the highest and lowest scores, respectively. The sample-sample contrastive loss is introduced to enhance the robustness of the

---

**Algorithm 1** Training the RL-based Teacher Selector

---

**Input**: Training dataset $\mathcal{D}$, a student model initialized as $\Theta^s = \Theta_0^s$, and a teacher selector initialized as $\theta = \theta_0$; hyper-parameters: $\lambda$, epoch number $L$, mini-batch size $b$ and learning rate $\beta$.

1: **for** epoch $l = 1$ to $L$ **do**
2:     Shuffle $\mathcal{D}$ to obtain a new training sequence.
3:     **for** each mini-batch $\mathcal{B} \in \mathcal{D}$ **do**
4:         Samples actions for each $q \in \mathcal{B}$ with the teacher selector to determine the selected teacher $T_{i^*}$ by:
5:             Compute the state $s_i$ for each teacher $T_i$ by Eq. 12;
6:             $i^* \leftarrow \arg\max_i \sigma(\mathbf{W}_i s_i + \mathbf{b}_i)$;
7:         Allocate the actions $a_i$ based on the selection;
8:         Stored $(s_i, a_i)$ to the episode history $\mathcal{H}$;
9:         Update the parameter $\Theta^s$ of the student model under the guidance of $T_{i^*}$ by Eq. 5.
10:     **for** each $(s_i, a_i) \in \mathcal{H}$ **do**
11:         Compute reward $r$:
12:             $r \leftarrow -\mathcal{L}_{\text{PR}} - \mathcal{L}_{\text{DL}}$;
13:         Compute the policy function $\pi_\theta(s_i, a_i)$ by Eq. 13;
14:         Update the parameter $\theta$ of the teacher selector by:
15:             $\theta \leftarrow \theta + \beta \sum_i r \sum_{q \in \mathcal{Q}} \nabla_\theta \pi_\theta(s_i, a_i)$.

---

vector representation, thereby promoting more stable training. We simplify its computation by directly classifying questions based on their respective datasets, as opposed to the semantic clustering approach employed in RouterDC:

$$\mathcal{L}_{\text{sample-sample}} = \sum_q -\log \frac{e^{\text{sim}\langle \mathcal{E}(q), \mathcal{E}(q^+) \rangle}}{e^{\text{sim}\langle \mathcal{E}(q), \mathcal{E}(q^+) \rangle} + \sum_{q^- \in \mathcal{Q}^-} e^{\text{sim}\langle \mathcal{E}(q), \mathcal{E}(q^-) \rangle}}, \tag{23}$$

where $q^+$ and $\mathcal{Q}^-$ denote an in-group question and an out-group set of the question $q$, respectively.

The training objective of the router consists of sample-LLM and sample-sample losses as:

$$\mathcal{L}_{\text{sim}} = \mathcal{L}_{\text{sample-LLM}} + \mathcal{L}_{\text{sample-sample}}. \tag{24}$$

We adopt mDeBERTaV3-base (He et al., 2021) as the language encoder to encode the input question. The dimension of the question embedding and the LLM embedding is 768. We train the model for 5000 epochs, using the AdamW (Loshchilov & Hutter, 2019) optimizer with a batch size of 16 and a learning rate of $2 \times 10^{-5}$.

### B.3 RL-BASED TEACHER SELECTION

Alg. 1 shows the training procedures of the RL-based teacher selector. During training, we alternately perform knowledge distillation and train the teacher selector. We regard the teacher selector as a broadly defined LLM router: it must simultaneously receive the question and the rationale, and it can be optimized within a unified framework together with the knowledge distillation process.

For training the teacher selector, we set the the epoch number $L$ to 2, the mini-batch size $b$ to 8, and the learning rate $\beta$ to $5 \times 10^{-5}$.

## C MORE EXPERIMENTAL DETAILS

### C.1 DETAILS OF MODEL SELECTION

In our experiments, we choose the small (77M), base (248M) and large (783M) model of the FLAN-T5 (Chung et al., 2024) series as student models. Due to computational resource constraint, we consider a multi-teacher ensemble comprising four LLMs: FLAN-T5 xlarge (2.85B), Llama 2-chat (Touvron et al., 2023)(7B), BioMistral-7B (Labrak et al., 2024), and Llama-3.1-8B-Instruct (Dubey et al., 2024).

We construct the teacher LLM ensemble purposefully: For the student models, FLAN-T5 xlarge serves as an homogeneous model with a larger number of parameters, while Llama 2-chat operates

Table 7: Statistics of the datasets we used in our experiments. The numbers represent the sample size of each partitions for each dataset.

| Domain | Dataset | Train | Test | Valid | Public |
|--------|---------|-------|------|-------|--------|
| ID | OBQA | 3965 | 500 | 500 | 992 |
| | ARC | 893 | 1165 | 295 | 224 |
| | Riddle | 2808 | 510 | 511 | 702 |
| | PQA | 400 | 400 | 100 | 100 |
| OOD | PIQA | 16113 | 919 | 919 | - |
| | BioASQ | 976 | 123 | 109 | - |

Table 8: Performance of ABKD for distilling the FLAN-T5 large model using different teacher LLMs.

| Teacher Model | OBQA | ARC | Riddle | PQA | Average |
|---------------|------|-----|--------|-----|---------|
| FLAN-T5 xlarge | 72.40 | 59.83 | 68.63 | 50.50 | 62.84 |
| Llama 2-chat | 70.00 | 58.45 | 67.84 | 51.25 | 61.89 |
| BioMistral-7B | 71.60 | 58.80 | 64.12 | 53.75 | 62.07 |
| Llama-3.1-8B-Instruct | 72.40 | 60.77 | 69.22 | 52.75 | 63.79 |

as a heterogeneous model, also with a higher parameter size. BioMistral-7B contributes domain-specific knowledge in biomedical reasoning, and Llama-3.1-8B-Instruct, as an updated version of Llama 2-chat, offers enhanced performance and establishes a selection tendency that facilitates the training of knowledge purification methods.

## C.2 DETAILS OF DATASET CONSTRUCTION

Our experiments involve six multiple choice question answering datasets, comprising four in-domain (ID) datasets: OpenBookQA (OBQA) (Mihaylov et al., 2018), AI2 Reasoning Challenge (ARC) (Clark et al., 2018), RiddleSense (Riddle), PubMedQA (PQA) (Jin et al., 2019) along with two out-of-domain (OOD) datasets: Physical Interaction Question Answering (PIQA) (Bisk et al., 2020) and BioASQ (Tsatsaronis et al., 2015).

For a fair comparison, we divide each in-domain dataset into four subsets: training, testing, evaluation, and public sets. The testing and the evaluation set inherit from the original dataset. For in-domain dataset, the training and the public sets are randomly partitioned from the training set of the original dataset in a ratio of 4:1. The public sets from each in-domain dataset collectively form a joint dataset comprising 2018 samples, used for training LLM routers. The remaining three subsets (training, testing and evaluation) are employed for knowledge distillation. Tab. 7 summarizes the division of both in-domain and out-of-domain datasets.

## C.3 DETAILS OF BASELINES

In our experiments, we compare the proposed methods against four baseline approaches, including **Inference**, which directly employs student model for evaluation; **Fine-tuning**, which fine-tunes the student model using the ground truth options as labels; **Distilling-Step-by-Step** (Hsieh et al., 2023), defined by Eq. 2, which leverages teacher's rationale as additional supervision; and **TinyLLM** (Tian et al., 2025), defined by Eq. 3, which integrates all rationales generated by teachers to train the student model. For the Distilling-Step-by-Step, we train with four teacher LLMs individually and report the best results for each dataset.

The standard knowledge distillation (Hinton et al., 2015) is not included in the baselines because, when applied to distilling LLMs, it merely replaces the ground truth labels with those generated by the teacher LLM. This process is functionally similar to fine-tuning, and its theoretical performance upper limit is lower than that of fine-tuning. Therefore, it is excluded from consideration.

We do not prioritize a comparative analysis between knowledge purification methods and single-teacher distillation approaches; instead, we emphasize the comparison with multi-teacher distilla-

tion methods. This preference stems from prior research (Hsieh et al., 2023) demonstrating the superior cross-task adaptability of multi-teacher distillation techniques. We additionally conduct an evaluation of the single-teacher method ABKD (Wang et al., 2025), which represents the state-of-the-art knowledge distillation method for LLM. ABKD uses a pair of $\alpha$-$\beta$ parameters to weight the divergence loss during the distillation stage to achieve a balance between Forward Kullback-Leibler Divergence (FKLD) and Reverse Kullback-Leibler Divergence (RKLD). In our experiment, we utilize ABKD to distill the FLAN-t5 large model using different teacher LLMs. The results are presented in Tab. 8. Notably, ABKD achieves an average accuracy up to 63.79%, outperforming the performance of Distilling-Step-by-Step yet remaining inferior to the optimal knowledge purification method (RL-based Teacher Selection), which attains an average accuracy of 67.55%.

(Xu et al., 2025) proposes a distillation framework TwT consists of two stages: Dual-Criteria Rejection Sampling and Habitual Reasoning Distillation. In the first stage, rationales produced by multi-teacher LLMs are screened. However, TwT is not regarded as a baseline method or as a knowledge-purification method in our paper, for three principal considerations: (1) The data screening in TwT can be interpreted as a combined approach that integrates a quality assessment process using LLMs with a resampling process based on similarity. This approach contrasts with our inclination to develop atomic methods within knowledge purification. In fact, the concept of introducing additional models for evaluation and performing selection based on similarity in TwT echoes the knowledge purification methods we propose. (2) TwT retains a pair of rationales from all rationales generated by multiple teacher LLMs, introducing new requirements for the subsequent distillation process, which does not align with the knowledge purification framework. (3) The quality evaluation of rationales in TwT relies on the weighting of multiple qualitative factors produced by LLM, which limits its generalization and real-time sampling capabilities. Nevertheless, the design motivation underlying TwT exhibits similarities with knowledge purification, and we anticipate to further exploring knowledge distillation in conjunction with the TwT framework in future work.

### C.4 REASONS FOR CHOOSING CMV AS METRIC

We quantitatively evaluate the effectiveness of knowledge purification methods in mitigating knowledge conflicts among teacher LLM by computing the *Conflict Mitigating Value* (CMV). The CMV serves as a performance-based metric rather than relying on information-theoretic measures at the rationale level. We have opted for the performance-based CMV for two principal reasons.

First, rationale-level metrics lack universality. Information-theoretic metrics, such as Jensen-Shannon Divergence, effectively quantify knowledge aggregation that produces new rationales. However, they are insufficient for methods like LLM routing, which require the selection of a single rationale from multiple options. Additionally, although we considered assessing the router's accuracy in choosing the rationale that aligns with the correct answer, this metric proves inadequate for measuring knowledge aggregation. Consequently, the pursuit of a unified rationale-level metric becomes particularly challenging.

Second, the rationales generated by teacher LLMs represent merely intermediate states within the knowledge distillation process and do not exert a direct influence on the performance of the final student model. Accordingly, we designed the CMV to concentrate on performance, enabling a more direct and meaningful evaluation.

## D SUPPLEMENTARY RESULTS

### D.1 EXTENDED EXPERIMENTS OF THE TINYLLM FRAMEWORK

TinyLLM (Tian et al., 2025) proposes a distillation paradigm that facilitates student model to learn from rationales generated by two teacher LLMs. Specifically, it adopts the following loss function when conduction multi-teacher knowledge distillation:

$$\mathcal{L}_{\text{TinyLLM}} = \mathcal{L}_{\text{PR}} + \sum_{j=1}^{n} \lambda_j \mathcal{L}_{\text{DL}\,j}, \tag{25}$$

where $\lambda_j$ is the importance weight for $T_j$.

Table 9: Extended experimental results of TinyLLM as the number of teacher LLMs increases from 1 to 4, where '+' denotes the addition of the specified LLM as a teacher, and $|\mathcal{T}|$ denotes the number of teacher LLMs participating in the distillation. The best results across different datasets are highlighted in **bold**.

| Student | Teacher Setting | $\|\mathcal{T}\|$ | OBQA | ARC | Riddle | PQA | Average |
|---|---|---|---|---|---|---|---|
| FLAN-T5 small 77M | FLAN-T5 xlarge | 1 | 40.00 | **30.56** | 42.75 | 48.50 | 40.45 |
| | + Llama 2-chat | 2 | **46.80** | 30.21 | 43.92 | 49.50 | **42.61** |
| | + BioMistral-7B | 3 | 40.80 | 30.47 | **44.31** | 46.75 | 40.58 |
| | + Llama-3.1-8B | 4 | 40.60 | **30.56** | 44.12 | **50.50** | 41.45 |
| FLAN-T5 base 248M | FLAN-T5 xlarge | 1 | 59.00 | 44.89 | 58.24 | **52.25** | 53.59 |
| | + Llama 2-chat | 2 | **63.40** | **46.44** | **60.98** | 50.50 | **55.33** |
| | + BioMistral-7B | 3 | 59.00 | 45.75 | 58.04 | 49.75 | 53.13 |
| | + Llama-3.1-8B | 4 | 58.80 | 46.18 | 58.82 | 47.25 | 52.76 |
| FLAN-T5 large 783M | FLAN-T5 xlarge | 1 | 71.60 | 59.48 | 68.43 | 50.50 | 62.50 |
| | + Llama 2-chat | 2 | **73.20** | 59.91 | **68.82** | 52.50 | **63.61** |
| | + BioMistral-7B | 3 | 69.20 | **60.26** | 66.08 | **54.50** | 62.51 |
| | + Llama-3.1-8B | 4 | 68.80 | 59.83 | 67.25 | 54.25 | 62.53 |

The basic TinyLLM framework relies on merely two teacher LLMs, constraining the practical application of knowledge distillation to improve the performance and domain-specific competencies of lightweight models. To explore the adaptability of TinyLLM to more teacher LLMs, we perform a series of extended experiments. We begin by using only the FLAN-T5 xlarge model as the teacher LLM and progressively incorporate additional teacher LLMs. In total, we conduct four groups of distillation training experiments with varying numbers of participating teachers. For a fair comparison, we set each $\lambda_j$ as 4, 2, 1.33, and 1 for the cases of 1, 2, 3, and 4 teachers, respectively. Tab. 9 presents the detailed results of these extended experiments.

Our observations indicate that as the number of teacher models further increases, the performance of the distilled student models actually declines. For all three student models, the best overall performance is achieved when two teacher LLMs participate in the distillation process. When the number of teacher LLMs reaches four, the performance of the distilled FLAN-T5 base model is even inferior to that achieved with a single teacher. These findings contradict our initial expectation that increasing the number of teachers would enhance knowledge diversity and generalization capabilities. We attribute this performance degradation to the emergence of knowledge conflicts among the teachers, emphasizing the critical need for knowledge purification.

### D.2 Supplementary Results of Adaptability toward Multiple Teacher LLMs

We adopt the same experimental setup in Appendix D.1 to evaluate the adaptability of knowledge purification methods in distillation training with a series of incremental teacher LLMs. Tab. 10~14 exhibit the complete results of the experiment.

### D.3 Generality toward Broader Task Domains and More Teacher LLMs

The current assessment utilizes four teacher LLMs, focusing mainly on the task domains of commonsense and biomedical reasoning. Our objective is to evaluate the generalization ability of knowledge purification methods across a broader range of task domains and more teacher LLMs. To this end, we extend our experiments using the MMLU dataset (Hendrycks et al., 2020), which encompasses 57 tasks from various branches of knowledge. Additionally, we introduce two supplementary teacher LLMs, llemma_7b (Azerbayev et al., 2023) and Mistral-7B-chat (Jiang et al., 2023), bringing a total of six teacher LLMs. In these experiments, we employ FLAN-T5 large as the student model for knowledge distillation and consider the similarity-based LLM routing for effective knowledge purification.

Tab. 15 demonstrates the results of evaluation. The similarity-based router achieves the highest average accuracy of 65.19%, surpassing the baselines by at least 7.3%. On the MMLU dataset, routing-based method also attains the highest accuracy of 55.26%. The results verifies the gener-

Table 10: Knowledge distillation results as the number of teacher LLMs increases from 1 to 4, applying **Knowledge Aggregation** for knowledge purification. The best results across different datasets are highlighted in **bold**.

| Student | Teacher Setting | $|\mathcal{T}|$ | OBQA | ARC | Riddle | PQA | Average |
|---|---|---|---|---|---|---|---|
| FLAN-T5 small 77M | FLAN-T5 xlarge | 1 | 40.00 | **30.56** | 42.75 | 48.50 | 40.45 |
| | + Llama 2-chat | 2 | 39.80 | 30.21 | 43.73 | 51.25 | 41.25 |
| | + BioMistral-7B | 3 | 40.20 | 30.47 | 43.14 | 49.50 | 40.83 |
| | + Llama-3.1-8B | 4 | **40.40** | 30.47 | **43.92** | **53.25** | **42.01** |
| FLAN-T5 base 248M | FLAN-T5 xlarge | 1 | **59.00** | 44.89 | 58.24 | **52.25** | **53.59** |
| | + Llama 2-chat | 2 | 55.60 | 45.92 | 57.06 | 50.50 | 52.27 |
| | + BioMistral-7B | 3 | 58.20 | **46.35** | **58.63** | 50.75 | 53.48 |
| | + Llama-3.1-8B | 4 | 58.00 | 45.75 | 58.43 | 51.50 | 53.42 |
| FLAN-T5 large 783M | FLAN-T5 xlarge | 1 | 71.60 | 59.48 | 68.43 | 50.50 | 62.50 |
| | + Llama 2-chat | 2 | 69.60 | 59.06 | 66.47 | 49.50 | 61.16 |
| | + BioMistral-7B | 3 | **72.00** | 58.80 | **69.02** | 52.50 | 63.08 |
| | + Llama-3.1-8B | 4 | 71.40 | **59.74** | 68.63 | **53.50** | **63.32** |

Table 11: Knowledge distillation results as the number of teacher LLMs increases from 1 to 4, applying **Plackett-Luce Ranking** for knowledge purification. The best results across different datasets are highlighted in **bold**.

| Student | Teacher Setting | $|\mathcal{T}|$ | OBQA | ARC | Riddle | PQA | Average |
|---|---|---|---|---|---|---|---|
| FLAN-T5 small 77M | FLAN-T5 xlarge | 1 | 40.00 | 30.56 | 42.75 | 48.50 | 40.45 |
| | + Llama 2-chat | 2 | 40.20 | 30.64 | 43.33 | 49.75 | 40.98 |
| | + BioMistral-7B | 3 | **42.20** | 30.73 | 44.31 | 50.25 | 41.87 |
| | + Llama-3.1-8B | 4 | 42.00 | **31.76** | **45.69** | **50.50** | **42.49** |
| FLAN-T5 base 248M | FLAN-T5 xlarge | 1 | 59.00 | 44.89 | 58.24 | 52.25 | 53.59 |
| | + Llama 2-chat | 2 | 60.20 | 45.41 | **58.82** | 52.75 | 54.30 |
| | + BioMistral-7B | 3 | **63.60** | 45.58 | 58.63 | 52.25 | 55.02 |
| | + Llama-3.1-8B | 4 | 62.40 | **46.27** | 58.63 | **54.75** | **55.51** |
| FLAN-T5 large 783M | FLAN-T5 xlarge | 1 | 71.60 | 59.48 | 68.43 | 50.50 | 62.50 |
| | + Llama 2-chat | 2 | 72.00 | **59.83** | 68.82 | 50.50 | 62.79 |
| | + BioMistral-7B | 3 | 71.80 | 59.66 | 69.02 | **56.50** | 64.25 |
| | + Llama-3.1-8B | 4 | **72.60** | 59.48 | **69.41** | 56.50 | **64.50** |

alization ability of the LLM routing method toward broader task domains and a large number of teacher LLMs, highlighting the effectiveness of the knowledge purification framework.

Despite our current inability to conduct experiments involving a greater number of teacher LLMs (8, 10, or more) due to computational resource limitations, we are encouraged by the success of existing LLM routing methods when applied to more than 10 candidate LLMs (Chen et al., 2024; Hu et al., 2024). It is evident that while knowledge purification methods improve performance, the effectiveness of knowledge distillation does not increase indefinitely with the addition of teacher LLMs. A more practical goal is to leverage an appropriate number of teacher models to enhance the overall capabilities and specific expertise of the student model.

Furthermore, We also look forward to validating the performance of knowledge purification across a wider range of NLP applications. This may necessitate stronger student models, as well as more sophisticated processes for distillation data sampling. We intend to explore these scenarios in our future research to further advance the field.

## D.4 CASE STUDY

We present a detailed case study and visualization of the knowledge purification process on the OBQA dataset, as illustrated in Fig. 5.

Table 12: Knowledge distillation results as the number of teacher LLMs increases from 1 to 4, applying **PLM Classifier** for knowledge purification. The best results across different datasets are highlighted in **bold**.

| Student | Teacher Setting | $|\mathcal{T}|$ | OBQA | ARC | Riddle | PQA | Average |
|---|---|---|---|---|---|---|---|
| FLAN-T5 small 77M | FLAN-T5 xlarge | 1 | 40.00 | 30.56 | 42.75 | 48.50 | 40.45 |
| | + Llama 2-chat | 2 | 40.80 | 30.21 | 45.69 | 51.75 | 42.11 |
| | + BioMistral-7B | 3 | 43.60 | 30.39 | 48.63 | 52.50 | 43.78 |
| | + Llama-3.1-8B | 4 | **44.20** | **30.64** | **49.22** | **53.75** | **44.45** |
| FLAN-T5 base 248M | FLAN-T5 xlarge | 1 | 59.00 | 44.89 | 58.24 | 52.25 | 53.59 |
| | + Llama 2-chat | 2 | 61.20 | 46.27 | 58.63 | 52.25 | 54.59 |
| | + BioMistral-7B | 3 | 61.00 | **46.44** | **59.41** | 52.00 | 54.71 |
| | + Llama-3.1-8B | 4 | **63.60** | 46.35 | 59.22 | **55.00** | **56.04** |
| FLAN-T5 large 783M | FLAN-T5 xlarge | 1 | 71.60 | 59.48 | 68.43 | 50.50 | 62.50 |
| | + Llama 2-chat | 2 | 72.40 | **59.83** | 68.82 | 51.75 | 63.20 |
| | + BioMistral-7B | 3 | 72.80 | 59.66 | 69.02 | 60.00 | 65.37 |
| | + Llama-3.1-8B | 4 | **74.20** | 59.48 | **69.41** | **62.50** | **66.40** |

Table 13: Knowledge distillation results as the number of teacher LLMs increases from 1 to 4, applying **Similarity-based Router** for knowledge purification. The best results across different datasets are highlighted in **bold**.

| Student | Teacher Setting | $|\mathcal{T}|$ | OBQA | ARC | Riddle | PQA | Average |
|---|---|---|---|---|---|---|---|
| FLAN-T5 small 77M | FLAN-T5 xlarge | 1 | 40.00 | 30.56 | 42.75 | 48.50 | 40.45 |
| | + Llama 2-chat | 2 | 42.00 | 30.47 | 46.08 | 49.75 | 42.08 |
| | + BioMistral-7B | 3 | 46.00 | 32.02 | 49.22 | 51.25 | 44.62 |
| | + Llama-3.1-8B | 4 | **48.60** | **32.19** | **49.61** | **52.25** | **45.66** |
| FLAN-T5 base 248M | FLAN-T5 xlarge | 1 | 59.00 | 44.89 | 58.24 | 52.25 | 53.59 |
| | + Llama 2-chat | 2 | 60.60 | 45.75 | 60.00 | 52.50 | 54.71 |
| | + BioMistral-7B | 3 | 63.80 | **46.35** | 60.20 | 53.00 | 55.84 |
| | + Llama-3.1-8B | 4 | **65.60** | **46.35** | **60.78** | **53.50** | **56.56** |
| FLAN-T5 large 783M | FLAN-T5 xlarge | 1 | 71.60 | 59.48 | 68.43 | 50.50 | 62.50 |
| | + Llama 2-chat | 2 | 74.00 | 59.83 | 69.41 | 54.25 | 64.37 |
| | + BioMistral-7B | 3 | 75.20 | 59.91 | 69.80 | **62.25** | 66.79 |
| | + Llama-3.1-8B | 4 | **76.60** | **60.60** | **70.59** | 61.00 | **67.20** |

# E  THE USE OF LARGE LANGUAGE MODELS

In this section, we clarify that no LLMs were employed during the original conceptualization, or drafting phases of this paper. All core content and findings are the result of original research and critical evaluation by the authors. The use of LLMs was strictly limited to post-drafting linguistic refinements, such as identifying grammatical errors, correcting typos, and improving sentences.

Table 14: Knowledge distillation results as the number of teacher LLMs increases from 1 to 4, applying **RL-based Teacher Selection** for knowledge purification. The best results across different datasets are highlighted in **bold**.

| Student | Teacher Setting | $|\mathcal{T}|$ | OBQA | ARC | Riddle | PQA | Average |
|---|---|---|---|---|---|---|---|
| FLAN-T5 small 77M | FLAN-T5 xlarge | 1 | 40.00 | 30.56 | 42.75 | 48.50 | 40.45 |
| | + Llama 2-chat | 2 | 41.40 | 30.73 | 47.84 | 48.75 | 42.18 |
| | + BioMistral-7B | 3 | 44.40 | **31.59** | 48.04 | 52.25 | 44.07 |
| | + Llama-3.1-8B | 4 | **46.80** | 30.39 | **48.82** | **52.50** | **44.63** |
| FLAN-T5 base 248M | FLAN-T5 xlarge | 1 | 59.00 | 44.89 | 58.24 | 52.25 | 53.59 |
| | + Llama 2-chat | 2 | 60.80 | 46.44 | 60.78 | 50.50 | 54.63 |
| | + BioMistral-7B | 3 | 63.60 | 46.61 | 60.20 | 51.75 | 55.54 |
| | + Llama-3.1-8B | 4 | **65.00** | **46.70** | **61.76** | **53.25** | **56.68** |
| FLAN-T5 large 783M | FLAN-T5 xlarge | 1 | 71.60 | 59.48 | 68.43 | 50.50 | 62.50 |
| | + Llama 2-chat | 2 | 73.60 | 60.60 | 69.02 | 51.25 | 63.62 |
| | + BioMistral-7B | 3 | 72.00 | 60.77 | 70.00 | 61.75 | 66.13 |
| | + Llama-3.1-8B | 4 | **75.20** | **61.12** | **70.39** | **63.50** | **67.55** |

Table 15: Supplementary results on commonsense reasoning, biomedical reasoning, and multitask language understanding with six teacher LLMs. The FLAN-T5 large model serves as the student model. The best results across different datasets are highlighted in **bold**.

| Method | OBQA | ARC | Riddle | PQA | MMLU | Average |
|---|---|---|---|---|---|---|
| Inference | 50.40 | 51.07 | 39.80 | 45.50 | 45.10 | 46.37 |
| Distilling-Step-by-Step | 71.60 | **60.77** | 68.43 | 51.00 | 51.84 | 60.73 |
| TinyLLM | 70.40 | 54.25 | 67.25 | 52.75 | 49.28 | 58.79 |
| Similarity-based Router | **77.00** | 60.17 | **70.78** | **62.75** | **55.26** | **65.19** |

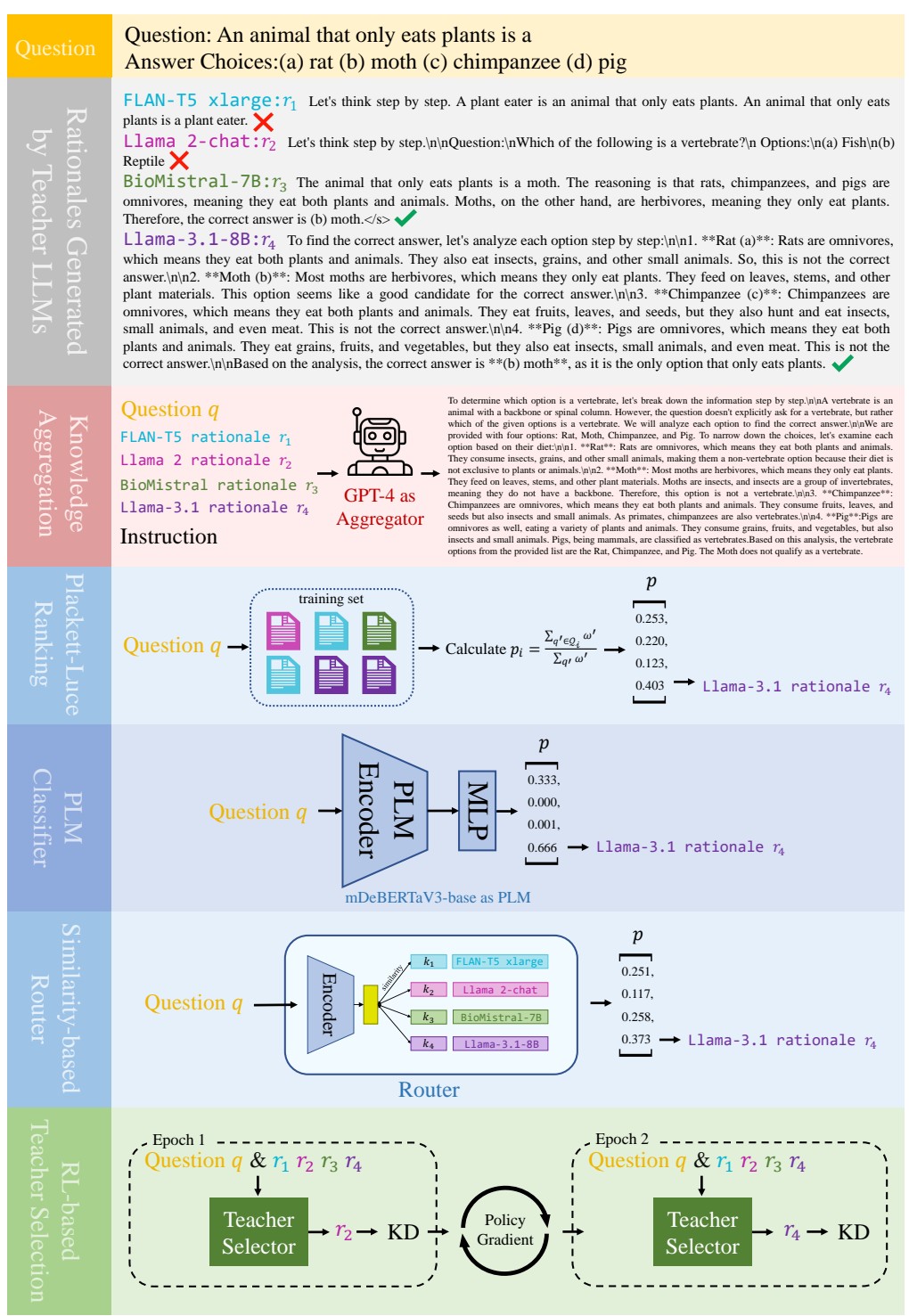

Figure 5: An example on the OBQA dataset. Five proposed methods are used for knowledge purification.

