# OpenReview forum: "Exploring Knowledge Purification in Multi-Teacher Knowledge Distillation for LLMs"
_ICLR.cc/2026/Conference — ICLR 2026 Poster_

### Official Review · Reviewer_Tdsi · 2025-10-23

**Soundness:** 3
**Presentation:** 3
**Contribution:** 3
**Rating:** 6
**Confidence:** 4

**Summary:**

This paper addresses a critical problem in multi-teacher knowledge distillation (KD) for Large Language Models (LLMs): the performance degradation that occurs as the number of teachers increases, attributed to knowledge conflicts. The authors introduce the concept of Knowledge Purification, which aims to consolidate the rationales from multiple teachers into a single and coherent rationale. They propose five purification methods in total, Knowledge Aggregation using an LLM to synthesize rationales, three LLM routing approaches (Plackett-Luce Ranking, PLM Classifier, and Similarity-based Router) which select the best single teacher's rationale based on the input question, and RL-based Teacher Selection that uses the student's performance as a reward to choose a teacher. Extensive experiments on commonsense and biomedical reasoning tasks show that routing-based and RL-based methods significantly outperform baselines like TinyLLM, and good generalization to out-of-domain datasets.

**Strengths:**

* The proposal of the concept of knowledge purification
* Five methods are proposed for a comparative analysis across multiple dimensions (performance, CMV, out-of-domain generalization, etc.), and the analysis (e.g., Table 2) shows the trade-offs of each method
* The proposed methods consistently outperform baselines, as well as the out-of-domain generalization

**Weaknesses:**

* The experiments are limited to four selected teacher models
* Though the overall performance gain is clear, why certain methods work better needs more explanation. It would be helpful if the authors could provide more analyses of pros and cons of each method
* Excluding the TwT baseline (in Appendix C.3) is reasonable but not fully convincing

**Questions:**

1. The performance of knowledge aggregation is relatively weak. Have you investigated why a powerful LLM fails to synthesize a good consolidated rationale? Is the issue the aggregation prompt or the task is inherently difficult?
2. Since the router-based methods perform well and they only need the input question, could it be used to select the teacher per query during the data generation?

---

> ### Author Response · Authors · 2025-11-16
> **Response to Reviewer Tdsi (1/2)**
>
> We sincerely appreciate your review and valuable feedback. We address your concerns point by point below.
>
> > **Q1:** The performance of knowledge aggregation is relatively weak. Have you investigated why a powerful LLM fails to synthesis a good consolidated rationale? Is the issue the aggregation prompt or the task is inherently difficult?
>
> We appreciate your insightful question. First, we rule out the selection of the aggregation prompt or the aggregator as reasons for the relatively weak performance. In Appendix B.1, we introduce the aggregation prompt, which has been optimized for multiple iterations. This prompt combines in-context examples with specific instructions designed to capture the key points of rationales. We tested various prompt forms; however, many failed to generate reasonable consolidated rationales, resulting in decreased distillation performance.
>
> Initially, we used GPT-4 as the aggregator. We supplemented our experiments with the relatively weaker Llama-3.1-70b as the aggregator while distilling the FLAN-t5 large student model. The results are shown below, indicating that using a more powerful model as the aggregator does not yield a significant performance improvement.
>
> | Aggregator | OBQA | ARC | Riddle | PQA | Avg. |
> |-|-|-|-|-|-|
> | Llama-3.1-70b | 71.2 | **60.8** | 67.7 | 52.5 | 63.0 |
> | GPT-4 | **71.4** | 59.7 | **68.6** | **53.5** | **63.3** |
>
> We attribute the performance disadvantage to **the lack of explicit guidance**. During knowledge aggregation, the consolidated rationale is generated without external supervision. Although the instructions are designed to guide the process, we cannot ensure that the aggregator effectively eliminates knowledge conflicts. Instead, we rely entirely on the model's inherent language processing capabilities, which may retain conflicts or reasoning paths associated with incorrect responses. Consequently, compared to routing-based methods and RL-based teacher selection which involve explicit training, knowledge aggregation exhibits notable performance disadvantages.
>
> > **Q2:** Since the router-based methods perform well and they only need the input question, could it be used to select the teacher per query during the data generation?
>
> Thanks for your meaningful question. Indeed, router-based approaches can be used for the teacher selection per query. In Section 5.4, we introduce this paradigm of utilizing LLM routers to guide knowledge distillation. This approach has demonstrated the ability to achieve excellent distillation results while effectively minimizing computational costs during the data sampling stage.
>
> > **W1:** The experiments are limited to four teacher models.
>
> Thanks for your meaningful feedback. We conduct additional experiments involving two more teacher LLMs, llemma_7b [1] and Mistral-7B-chat [2] (6 teachers in total). We employ FLAN-T5 large is as the student model for knowledge distillation and utilize the similarity-based LLM routing for knowledge purification. The results are shown in the table below, which demonstrates that the knowledge purification method continues to show performance advantages over the baselines, with 6 teacher LLMs.
>
> | Method | OBQA | ARC | Riddle | PQA | Avg. |
> |-|-|-|-|-|-|
> | Inference | 50.4 | 51.1 | 39.8 | 45.5 | 46.7 |
> | Distilling-Step-by-Step | 71.6 | **60.8** | 68.4 | 51.0 | 63.0 |
> | TinyLLM | 70.4 | 54.3 | 67.3 | 52.8 | 61.2 |
> | Similarity-based Router | **77.0** | 60.2 | **70.8** | **62.8** | **67.7** |
>
> It is evident that while knowledge purification enhances performance, the efficacy of knowledge distillation does not increase infinitely with the number of teacher LLMs. A more practical goal is to leverage an appropriate number of teacher models to enhance the overall capabilities and specific expertise of the student model. Within the constraints of computational resources, our experiments confirm the superiority of the knowledge purification method compared to existing approaches with 6 teacher LLMs. For the scenario with more teacher models, the strong performance of existing LLM routing methods in LLM pool with over 10 LLMs [3] is encouraging and we look forward to further integrating teacher models in our future research.

---

> > ### Author Response · Authors · 2025-11-16
> > **Response to Reviewer Tdsi (2/2)**
> >
> > > **W2:** Though the overall performance gain is clear, why certain methods work better needs more explanation. It would be helpful if the authors could provide more analyses of pros and cons of each method.
> >
> > We appreciate your valuable suggestion and we summarize the pros and cons of each method as follows:
> >
> > - **Knowledge Aggregation** serves as a intuitive method of knowledge purification. However, the experiments verify that its improvement in distillation is limited, and the introduction of the aggregator increases the computational cost.
> > - **LLM Routing** facilitates the distillation process through query-specific LLM selection, demonstrating strong performance across a wide range of datasets and student model settings. Among the various routers, the similarity-based router achieves the best average performance. This approach to knowledge purification enables dynamic and lightweight knowledge distillation.
> > - **RL-based Teacher Selection** delivers the highest distillation performance, owing to the optimization of the reward-guided selector. However, the iterative training incurs significant cost and extended time, which may limit its generalization capacity.
> >
> > > **W3:** Excluding the TwT baseline (in Appendix C.3) is reasonable but not fully convincing.
> >
> > We exclude TwT from the baselines and the knowledge purification methods mainly because its data sampling can be regarded as a combination of multiple sampling tricks, and it needs to retain a pair of rationales which is contrary to the defination of knowledge purification. Nevertheless, we are pleased to present the results of TwT distillation with FLAN-t5 large as the student model. The results are shown below.
> >
> > | Method | OBQA | ARC | Riddle | PQA | Avg. |
> > |-|-|-|-|-|-|
> > | DCRS | 72.4 | 58.6 | 69.0 | 54.5 | 63.6 |
> > | TwT (DCRS + HaRD) | 76.2 | 60.8 | **71.8** | 59.3 | 67.0 |
> > | Similarity-based Router | **76.6** | 60.6 | 70.6 | 61.0 | 67.2 |
> > | Teacher Selection | 75.2 | **61.1** | 70.4 | **63.5** | **67.6** |
> >
> > It can be seen that TwT achieves relatively ideal results, but this is based on its combination of rejection sampling and a three-stage distillation training with high computational consumption. By comparison, the knowledge purification methods achieve performance comparable to or even surpassing it through just one distillation training.
> >
> > ## References
> >
> > [1] Azerbayev, Z. et, al. (2023) Llemma: An Open Language Model For Mathematics. *arXiv:2310.10631.*
> > [2] Jiang, A. et, al. (2023) Mistral 7b. *arXiv: 2310.06825.*
> > [3] <https://github.com/withmartian/routerbench>
> > [4] Xu, J., et al. (2025). Twt: Thinking without tokens by habitual reasoning distillation with multi-teachers' guidance. *arXiv:2503.24198.*
> >
> > We look forward to your feedback!

---

> > > ### Author Response · Authors · 2025-11-28
> > > **We hope to receive your feedback.**
> > >
> > > Dear Reviewer Tdsi,
> > >
> > > We would like to express our gratitude for your valuable comments on our submission. As the discussion phase is approaching its end, we would like to kindly confirm whether we have sufficiently addressed all of your concerns (or at least part of them). Should there be any remaining questions or areas requiring further clarification, please do not hesitate to let us know. If you are satisfied with our responses, we would greatly appreciate your consideration in adjusting the evaluation scores accordingly. Thank you for your hard work and support.
> > >
> > > Best regards,
> > >
> > > The author of Submission13324

---

### Official Review · Reviewer_Wb1Y · 2025-10-30

**Soundness:** 2
**Presentation:** 3
**Contribution:** 2
**Rating:** 4
**Confidence:** 3

**Summary:**

This work addresses multi-teacher knowledge distillation (KD), focusing on the challenges of knowledge conflict (i.e.divergent rationales among teachers) and computational cost when aggregating many teacher models. The authors propose Knowledge Purification, a concept for consolidating the rationales from multiple teacher LLMs into a single rationale to use for distillation. They design five purification methods based on aggregation, routing, and RL-based teacher selection styles.

**Strengths:**

- The introduction of Knowledge Purification reframes multi-teacher distillation from the perspective of rationale integration rather than mere logit or feature averaging, addressing an important problem of multi-teacher KD.

- The paper proposes five distinct purification strategies (aggregation, routing, and RL-based).

- The paper is well written and structured.

**Weaknesses:**

- The experimental evaluation is limited to commonsense and biomedical reasoning datasets. To establish the generality of the proposed approach, it should be extended to a broader range of tasks such as mathematical reasoning, coding, and instruction following.

- Although the paper emphasizes improved efficiency, it does not provide quantitative evidence (e.g., training time, GPU hours) compared to existing multi-teacher methods like TinyLLM or TwT.

- The baselines are limited to step-by-step distillation and TinyLLM, omitting state-of-the-art knowledge distillation approaches such as ABKD, MiniLLM, DistillM, or CKA-KD, which could challenge the claimed improvements.

**Questions:**

See weaknesses section

---

> ### Author Response · Authors · 2025-11-16
> **Response to Reviewer Wb1Y**
>
> We sincerely appreciate your review and valuable feedback. We address your concerns point by point below.
>
> > **Q1:** The experimental evaluation is limited to commonsense and biomedical reasoning datasets. To establish the generality of the proposed approach, it should be extended to a broader range of tasks such as mathematical reasoning, coding, and instruction following.
>
> We appreciate your insightful feedback. We conduct supplementary experiments on the MMLU dataset [1], which covers 57 tasks from various branches of knowledge. In these experiments, we additionally introduce two teacher LLMs-llemma_7b [2] and Mistral-7B-chat [3]-and employ FLAN-T5 large as the student model for knowledge distillation. We consider the similarity-based LLM routing for knowledge purification and the results on MMLU are shown in the table below.
>
> | Method | MMLU |
> |-|-|
> | Inference | 45.1 |
> | Distilling-Step-by-Step | 51.8 |
> | TinyLLM | 49.3 |
> | Similarity-based Router | **55.3** |
>
> The experimental results on the multitask dataset demonstrate the superiority of knowledge purification. We look forward to validating its effectiveness across a wider range of NLP applications, which will require stronger student model configurations and resampling of the distillation data. We plan to explore these scenarios in our future work.
>
> > **Q2:** Although the paper emphasizes improved efficiency, it does not provide quantitative evidence (e.g., training time, GPU hours) compared to existing multi-teacher methods like TinyLLM or TwT.
>
> Thanks for your meaningful suggestion. Using the GPU hours trained on the NVIDIA A100 GPUs for statistics, we assess the training efficiency of different methods when distilling the FLAN-T5 large model on the ARC dataset. For knowledge purification methods, we consider the computational consumption of both the purification stage (e.g., aggregation, training the router) and the distillation stage. The results are shown in the table below.
>
> | Method | Purification Stage (GPU-hour) | Distillation Stage (GPU-hour) |
> |-|-|-|
> | Distilling-Step-by-Step | - | 1.1 (per teacher) |
> | TinyLLM | - | 2.6 |
> | Knowledge Aggregation | 5.2 (for open-source aggregator Llama-3.1-70b) | 1.5 |
> | Plackett-Luce Ranking | 0.1 | 1.3 |
> | PLMClassifier | 0.5 | 1.2 |
> | Similarity-based Router | 0.6 | 1.2 |
> | Teacher Selection | - | 3.5 |
>
> It shows that knowledge purification methods enhance the efficiency of the distillation stage compared to TinyLLM, which employs all rationales. Notably, training the LLM router demands fewer computational resources than the distillation process. Furthermore, given the generalization capabilities of routers, the impact of knowledge purification on improving efficiency will be more pronounced when conducting knowledge distillation on out-of-domain data.
>
> > **Q3:** The baselines are limited to step-by-step distillation and TinyLLM, omitting state-of-the-art knowledge distillation approaches, which could challenge the claimed improvements.
>
> We appreciate your insightful feedback. During our research, we surveyed state-of-the-art knowledge distillation methods, including ABKD, MiniLLM, and DISTILLM. We chose not to include these methods in our baselines because they are designed for single-teacher distillation, while multi-teacher distillation has been shown to outperform them, particularly in cross-domain tasks. Therefore, we focused our evaluations on comparing the knowledge purification method with a multi-teacher distillation baseline.
>
> Nevertheless, we are pleased to present the results of ABKD [4] with FLAN-t5 large as the student model. The ABKD method uses a pair of $\alpha$-$\beta$ parameters to weight the divergence loss during the distillation stage to achieve a balance between FKLD and RKLD, which is the state-of-the-art method, surpassing MiniLLM and DISTILLM. We optimized the $\alpha$-$\beta$ selections for each teacher LLM and retained the optimal results as follows.
>
> | Teacher Model | OBQA | ARC | Riddle | PQA | Avg. |
> |-|-|-|-|-|-|
> | FLAN-T5 xlarge | **72.4** | 59.8 | 68.6 | 50.5 | 62.8 |
> | Llama 2-chat | 70.0 | 58.5 | 67.8 | 51.3 | 61.9 |
> | BioMistral-7B | 71.6 | 58.8 | 64.1 | **53.8** | 62.1 |
> | Llama-3.1-8B-Instruct | **72.4** | **60.8** | **69.2** | 52.8 | **63.8** |
>
> While ABKD demonstrates better performance than the Distilling-Step-by-Step method, it still lags behind the best-performing knowledge purification method (RL-based Teacher Selection), which achieves an average accuracy of **67.55**.
>
> ## References
>
> [1] <https://huggingface.co/datasets/cais/mmlu>
> [2] Azerbayev, Z. et, al. (2023) Llemma: An Open Language Model For Mathematics. *arXiv:2310.10631.*
> [3] Jiang, A. et, al. (2023) Mistral 7b. *arXiv: 2310.06825.*
> [4] Wang, G. et, al. (2025) ABKD: Pursuing a Proper Allocation of the Probability Mass in Knowledge Distillation via $\alpha$-$\beta$-Divergence. *ICML 2025.*
>
> We look forward to your feedback!

---

> > ### Author Response · Authors · 2025-11-28
> > **We hope to receive your feedback.**
> >
> > Dear Reviewer Wb1Y,
> >
> > We would like to express our gratitude for your valuable comments on our submission. As the discussion phase is approaching its end, we would like to kindly confirm whether we have sufficiently addressed all of your concerns (or at least part of them). Should there be any remaining questions or areas requiring further clarification, please do not hesitate to let us know. If you are satisfied with our responses, we would greatly appreciate your consideration in adjusting the evaluation scores accordingly. Thank you for your hard work and support.
> >
> > Best regards,
> >
> > The author of Submission13324

---

### Official Review · Reviewer_Bt3j · 2025-11-01

**Soundness:** 3
**Presentation:** 3
**Contribution:** 2
**Rating:** 6
**Confidence:** 3

**Summary:**

This paper addresses a critical challenge in multi-teacher knowledge distillation (MTKD) for Large Language Models (LLMs): the performance degradation caused by knowledge conflicts among the rationales provided by multiple teacher models. The authors identify that simply increasing the number of teachers in frameworks like TinyLLM does not monotonically improve student performance, often harming it due to conflicting or hallucinated reasoning paths. To solve this, the authors introduce the concept of "Knowledge Purification" (KP), which aims to consolidate the rationales from multiple teachers into a single, coherent rationale before distillation. This process mitigates conflicts and provides the student model with a unified source of knowledge. Through extensive experiments on commonsense and biomedical reasoning tasks, the paper demonstrates that KP methods, particularly the Similarity-based Router and RL-based Teacher Selection, consistently outperform strong baselines like TinyLLM and Distilling-Step-by-Step.

**Strengths:**

1. The idea of "Knowledge Purification" is a direct, intuitive, and novel solution to the clearly identified problem of knowledge conflict in MTKD.
2. The proposal of five methods from different families (aggregation, routing, RL) provides a thorough exploration of the solution space. This allows for a nuanced comparison of trade-offs between performance, computational cost, and transferability.

**Weaknesses:**

1. As acknowledged in the limitations, the study is constrained to a ensemble of only four teacher LLMs. A critical question remains: how do these methods scale to 10, 20, or even more teachers? While the results with 4 teachers are promising, the effectiveness and computational overhead of, for example, the RL-based method or the PL ranking with a much larger pool of teachers is unexplored.
2. The study is exclusively validated on multiple-choice question answering tasks. While this is a standard benchmark for reasoning, the generality of Knowledge Purification to other NLP tasks like open-ended generation, summarization, or translation is not established.
3. While the routers offer excellent inference-time efficiency, the cost of training them is non-trivial (requiring a "public set" and 5000 training epochs). A discussion on the trade-off between the cost of training a router versus the cost of repeatedly sampling from all teachers for distillation across multiple tasks would be beneficial.

**Questions:**

N/A

---

> ### Author Response · Authors · 2025-11-16
> **Response to Reviewer Bt3j (1/2)**
>
> We sincerely appreciate your review and valuable feedback. We address your concerns point by point below.
>
> > **Q1:** The study is constrained to a ensemble of only four teacher LLMs. How do these methods scale to 10, 20, or even more teachers? While the results with 4 teachers are promising, the effectiveness and computational overhead with a much larger pool of teachers is unexplored.
>
> We appreciate your insightful feedback. We conduct additional experiments involving two more teacher LLMs, llemma_7b [1] and Mistral-7B-chat [2] (6 teachers in total). We employ FLAN-T5 large is as the student model for multi-teacher knowledge distillation. We utilize the similarity-based LLM routing for knowledge purification and perform supplementary experiments on the MMLU dataset. The results are shown in the table below, which demonstrates that the knowledge purification method continues to show performance advantages over the baselines, with 6 teacher LLMs.
>
> | Method | OBQA | ARC | Riddle | PQA | MMLU |
> |-|-|-|-|-|-|
> | Inference | 50.4 | 51.1 | 39.8 | 45.5 | 45.1 |
> | Distilling-Step-by-Step | 71.6 | **60.8** | 68.4 | 51.0 | 51.8 |
> | TinyLLM | 70.4 | 54.3 | 67.3 | 52.8 | 49.3 |
> | Similarity-based Router | **77.0** | 60.2 | **70.8** | **62.8** | **55.3** |
>
> It is evident that while knowledge purification enhances performance, the efficacy of knowledge distillation does not increase infinitely with the number of teacher LLMs. A more practical goal is to leverage an appropriate number of teacher models to enhance the overall capabilities and specific expertise of the student model. Within the constraints of computational resources, our experiments confirm the superiority of the knowledge purification method compared to existing approaches with 6 teacher LLMs, addressing areas such as commonsense reasoning, biomedical reasoning, and multitask language understanding. For the scenario with more teacher models, the strong performance of existing LLM routing methods in LLM pool with over 10 LLMs is encouraging. We look forward to further integrating teacher models in our future research.
>
> > **Q2:** While this is a standard benchmark for reasoning, the generality of Knowledge Purification to other NLP tasks like open-ended generation, summarization, or translation is not established.
>
> Thanks for your meaningful suggestion. In our experimental verification, we focused primarily on multiple-choice QA tasks. On one hand, the response format of "rationale + answer" in multiple-choice QA task can be effectively optimized through knowledge distillation, as demonstrated by previous work [3]. On the other hand, for tasks such as open-ended generation and summarization, the capabilities of the student model become the main limiting factor in the effectiveness of the distillation. In our experiments, the selected student models—particularly the FLAN-T5 small and base models with fewer parameters—exhibit limited improvement in their performance on more open-ended tasks through distillation. Thus, validating knowledge purification methods in these tasks remains challenging.
>
> We believe knowledge purification can be generally applied to other NLP tasks due to its flexible framework, which doesn't overly restrict models or distillation techniques. We aim to verify its effectiveness across a wider range of NLP applications. To achieve this requires stronger student models and the data resampling process. However, due to computational resource constraints, we cannot conduct these validations under our current conditions, and we plan to reserve this for future work. Thank you again for your insightful feedback.

---

> > ### Author Response · Authors · 2025-11-16
> > **Response to Reviewer Bt3j (2/2)**
> >
> > > **Q3:** While the routers offer excellent inference-time efficiency, the cost of training them is non-trivial. A discussion on the trade-off between the cost of training a router versus the cost of repeatedly sampling from all teachers for distillation across multiple tasks would be beneficial.
> >
> > We appreciate your valuable suggestion. Using the FLAN-T5 large model for distillation training on the ARC dataset as an example, we record the GPU hour utilized by different methods to assess their efficiency. The results are shown in the table below.
> >
> > | Method | Purification Stage (GPU-hour) | Distillation Stage (GPU-hour) |
> > |-|-|-|
> > | Distilling-Step-by-Step | - | 1.1 (per teacher) |
> > | TinyLLM | - | 2.6 |
> > | Knowledge Aggregation | 5.2 (for open-source aggregator Llama-3.1-70b) | 1.5 |
> > | Plackett-Luce Ranking | 0.1 | 1.3 |
> > | PLMClassifier | 0.5 | 1.2 |
> > | Similarity-based Router | 0.6 | 1.2 |
> > | Teacher Selection | - | 3.5 |
> >
> > The knowledge purification methods enhance the efficiency of the distillation compared to TinyLLM, while training the LLM router demands fewer computational resources than the distillation process. Furthermore, given the generalization capabilities of routers, the impact of knowledge purification on improving efficiency will be more pronounced when conducting knowledge distillation on out-of-domain data.
> >
> > ## References
> >
> > [1] Azerbayev, Z. et, al. (2023) Llemma: An Open Language Model For Mathematics. *arXiv:2310.10631.*
> > [2] Jiang, A. et, al. (2023) Mistral 7b. *arXiv: 2310.06825.*
> > [3] Hsieh, C. et al. (2023) Distilling step-by-step! outperforming larger language models with less training data and smaller model sizes. *ACL Findings 2023.*
> >
> > We look forward to your feedback!

---

> > > ### Author Response · Authors · 2025-11-28
> > > **We hope to receive your feedback.**
> > >
> > > Dear Reviewer Bt3j,
> > >
> > > We would like to express our gratitude for your valuable comments on our submission. As the discussion phase is approaching its end, we would like to kindly confirm whether we have sufficiently addressed all of your concerns (or at least part of them). Should there be any remaining questions or areas requiring further clarification, please do not hesitate to let us know. If you are satisfied with our responses, we would greatly appreciate your consideration in adjusting the evaluation scores accordingly. Thank you for your hard work and support.
> > >
> > > Best regards,
> > >
> > > The author of Submission13324

---

### Official Review · Reviewer_EKzP · 2025-11-11

**Soundness:** 2
**Presentation:** 2
**Contribution:** 3
**Rating:** 4
**Confidence:** 3

**Summary:**

The paper proposes knowledge purification, which consolidates rationales from multiple teacher LLMs into a single rationale to address knowledge conflicts in distillation. Five methods are introduced: knowledge aggregation, LLM routing (Plackett-Luce ranking, PLM classifier, similarity-based router), and RL-based teacher selection. Experiments on commonsense and biomedical reasoning tasks show some performance gains, but the improvements are modest and lack groundbreaking insights.

**Strengths:**

(1) The focus on knowledge conflicts in multi-teacher distillation is attractive.

(2) Experiments cover multiple datasets and student models, providing a broad assessment.

(3) The five purification approaches offer varied perspectives, from simple aggregation to learned routing. Experimental results directly show the performance of the methods.

**Weaknesses:**

(1) The core idea of rationale consolidation resembles prior work on knowledge fusion and ensemble distillation. Methods like Plackett-Luce ranking and similarity-based routing are direct adaptations from the existing literature, showing limited substantial innovation. The paper only provide an overall view of the different methods.

(2) Methods like RL-based selection and aggregation involve significant complexity and does not present an obvious theoretical advantage through classical methods.

(3) While the experimental results show some improvements, the gains are often marginal (e.g., ~1–3% accuracy boosts in Table 1).

**Questions:**

1. How does knowledge purification fundamentally differ from applying ensemble methods (e.g., weighted averaging, majority voting or averaging student model) to the teacher rationales?

2. Why CMV was preferred over the information-theoretic metrics like Jensen-Shannon Divergence (JSD), which could quantify the divergence between teacher rationales?

3. Based on the experiments, which methods should we choose when facing the situation of multiple teacher distillation (under different scenarios)?

---

> ### Author Response · Authors · 2025-11-16
> **Response to Reviewer EKzP**
>
> We sincerely appreciate your review and valuable feedback. We address your concerns point by point below.
>
> > **Q1:** How does knowledge purification fundamentally differ from applying ensemble methods to the teacher rationales?
>
> Thanks for your meaningful question. The fundamental distinction between knowledge purification and ensembling methods lies in the focus on actively eliminating knowledge conflicts. In our paper, we define knowledge conflicts as the contradictory rationales generated by different teachers, influenced by factors like domain knowledge, reasoning paradigms, and hallucinations. We examine the limitations of TinyLLM [1], which is a representative method in distillation using weighted averaging, when applied to a larger number of teacher models (Appendix D.1). Furthermore, we introduce the concept of knowledge purification, which aims to actively eliminate these knowledge conflicts, and verify its enhancing effects on multi-teacher knowledge distillation through empirical experiments.
>
> > **Q2:** Why CMV was preferred over the information-theoretic metrics like JSD, which could quantify the divergence between teacher rationales?
>
> We appreciate your insightful question. We chose performance-based CMV over information-theoretic metrics for two main reasons. 1. Our initial goal was to design a rationale-level metric. While information-theoretic metrics like JSD are suitable for measuring knowledge aggregation, they fall short for methods like routing that select one from multiple rationales. We also considered the "Hit Rate" to assess whether the router selects the rationale correspond to the correct answer, but it does not apply to knowledge aggregation. Consequently, finding a unified rationale-level metric is challenging. 2. The rationales generated by teachers are merely intermediate states in knowledge distillation and do not directly influence the performance of the final student model. Therefore, we designed the CMV to focus on performance for a more direct evaluation.
>
> > **Q3:** Which methods should we choose when facing the situation of multiple teacher distillation?
>
> Thanks for your question. Our experimental verification indicates that knowledge purification methods utilizing LLM routing (e.g., similarity-based router) effectively balance both performance and efficiency. We also present the paradigm of using LLM routers to guide knowledge distillation in Section 5.4. As a result, we consider LLM routing to be the primary option for implementing knowledge purification.
>
> > **W1 & W2:** The core idea of rationale consolidation resembles prior work on knowledge fusion and ensemble distillation. Methods like PL ranking and similarity-based routing are direct adaptations from the existing literature. The paper only provide an overall view of the different methods. Methods like RL-based selection and aggregation involve significant complexity and does not present an obvious theoretical advantage through classical methods.
>
> We appreciate your insightful feedback. While we acknowledge that the core concepts of rationale consolidation similarities to established methods in knowledge fusion, research focusing on consolidation within the context of knowledge distillation—particularly concerning knowledge transfer among language models—is quite limited. To address this gap, we designed methods for knowledge purification from multiple perspectives, including relatively direct knowledge aggregation and RL-based selection, which uniformly optimizes purification and distillation.
>
> The primary contribution of our work lies not merely in the individual methods themselves but in the establishment of a comprehensive knowledge purification framework, which has been empirically validated to demonstrate positive impact on multi-teacher knowledge distillation. We believe this holistic approach offers significant advancements over traditional multi-teacher distillation techniques, particularly in enhancing the effectiveness of knowledge transfer.
>
> > **W3:** While the experimental results show some improvements, the gains are often marginal.
>
> We appreciate your feedback and acknowledge that the observed improvements may appear modest. However, these gains are significant within the context of state-of-the-art multi-teacher knowledge distillation methods. The consistent performance enhancements across various student models and tasks highlight the universal applicability of our knowledge purification framework. Furthermore, our results demonstrate its effectiveness in enhancing the robustness and reliability of multi-teacher knowledge distillation. We believe that the cumulative impact of these improvements, alongside the framework's empirical validation, merits consideration.
>
> ## References
>
> [1] Tian, Y. et, al. (2025) Beyond answers: Transferring reasoning capabilities to smaller llms using multi-teacher knowledge distillation. *WSDM 2025.*
>
> We look forward to your feedback!

---

> > ### Author Response · Authors · 2025-11-28
> > **We hope to receive your feedback.**
> >
> > Dear Reviewer EKzP,
> >
> > We would like to express our gratitude for your valuable comments on our submission. As the discussion phase is approaching its end, we would like to kindly confirm whether we have sufficiently addressed all of your concerns (or at least part of them). Should there be any remaining questions or areas requiring further clarification, please do not hesitate to let us know. If you are satisfied with our responses, we would greatly appreciate your consideration in adjusting the evaluation scores accordingly. Thank you for your hard work and support.
> >
> > Best regards,
> >
> > The author of Submission13324

---

### Author Response · Authors · 2025-11-26
**Follow-up on Submission #13324 regarding Reviewers**

Dear ICLR 2026 **AC**, **SAC**, and **PC**,

We would like to express our sincere gratitude to all the reviewers for their valuable feedback. We have carefully provided detailed clarifications and updated our submission accordingly.

However, we have not yet received responses from **Reviewer Tdsi**, **Reviewer Wb1Y**, **Reviewer Bt3j**, and **Reviewer EKzP**. We kindly request your assistance in reaching out to these reviewers. It would be greatly appreciated if you could encourage them to review our rebuttal, as we remain eager to ensure that their questions and concerns are fully addressed before the final evaluation.

We believe that constructive and timely communication between reviewers and authors is essential for the benefit of both parties. This would greatly help in ensuring a fair and informed final decision.

Thank you for your support.

Best regards,

The authors of Submission #13324

---

### Author Response · Authors · 2025-12-01
**Summary of Rebuttal**

We thank reviewers (Tdsi, Wb1Y, Bt3j, EKzP) and AC for their constructive feedback and time. We appreciate the unanimously recognition of **Knowledge Purification** as **a novel and intuitive solution for knowledge conflicts in MTKD**. Below, we summarize the key points of our rebuttal, highlighting the main efforts and additional experiments we conducted during the rebuttal period.

## 1. Expansion of Teacher LLMs and Task Generalization

A primary concern was whether the proposed knowledge purification methods are limited to the number of teacher models (Reviewers Tdsi, Bt3j) or the task domain (Reviewer Wb1Y). To support our claim of generality, we extended the experiments:

- **Additional Teacher LLMs**: We additionally incorporated llemma_7b and Mistral-7B-chat, with a total of **six** teacher LLMs.
- **New Task Domain**: MMLU dataset, which covers 57 tasks from various branches of knowledge.
- **Results**: The similarity-based LLM routing for knowledge purification continues to show performance advantages over the baselines (e.g., Distilling-Step-by-Step, TinyLLM) with 6 teacher LLMs.
   * **MMLU Performance**: Similarity-based Router (**55.3**) surpassed Distilling-Step-by-Step (51.8) and TinyLLM (49.3).
- **Conclusion**: It can be foreseen that the efficacy of knowledge distillation will not increase infinitely with the number of teacher LLMs. A more practical goal is to leverage an appropriate number of teacher models to enhance the overall capabilities and specific expertise of the student model.

## 2. Efficiency Analysis

Reviewers (Wb1Y, Bt3j) suggested quantitative evidence would be valuable in demonstrating the efficiency improvement of the knowledge purification methods. In response, we provided an analysis of the computational efficiency of proposed methods:

- **Experimental Setting**: We evaluated the training efficiency of different methods for distilling the FLAN-T5 large model on the ARC dataset, using GPU hours as the metric. For knowledge purification methods, we included both the purification stage (e.g., aggregation, training the router) and the distillation stage.
- **Results**: All methods enhanced distillation efficiency compared to TinyLLM, which employs all rationales.
   * **Knowledge Aggregation**: demonstrated the highest comsumpution due to the extensive call of the aggregator.
   * **RL-based Teacher Selection**: entailed considerable computational consumption for its iterative training process.
   * **LLM-Routing**: was proved to be the most efficient approach, requiring fewer computational resources for training the LLM router than for the distillation process.
- **Out-of-domain Training**: Given the generalization capabilities of routers, the impact of knowledge purification on improving efficiency will be more pronounced when conducting knowledge distillation on out-of-domain data.

## 3. Comprehensive Baseline Comparisons

Reviewers (Tdsi, Wb1Y) expressed concerns regarding the inclusion of certain baselines. In response, we broadened our baseline comparisons:

- **TwT**: We initially excluded TwT due to its reliance on multiple sampling techniques and the retention of pairs of rationales, which conflicts with the principles of knowledge purification.
   * **Supplementary Results**: TwT achieved ideal results but with high computational consumption. By comparison, the knowledge purification methods achieved performance comparable to, or even exceeding, that of TwT with only one distillation training.
- **SOTA Knowledge Distillation Methods**: We originally focused our evaluations on comparing knowledge purification methods with a multi-teacher baseline. We supplemented experiments with the SOTA single-teacher method **ABKD**.
   * **Supplementary Results**: The performance of ABKD (63.79) was inferior to that of the best-performing knowledge purification method (**67.55**), validating the superiority of our framework.

## 4. Methodological Insights

To address concerns about the theoretical and practical effectiveness of our methods (Reviewers Tdsi, EKzP), we explored several distinct aspects:

- **Limitation of Knowledge Aggregation**: We acknowledged the limitation of knowledge aggregation, noting that its lack of **explicit guidance** in the consolidation process often led to suboptimal distillation results.
- **Comparison to Ensemble Methods**: We clarified the distinction between knowledge purification and model ensembling methods, emphasizing that our framework actively addresses **knowledge conflicts** between teacher models rather than simply weighting their outputs.

## Addtional Clarifications

We also provide analysis on *knowledge conflicts mitigating metrics* (Reviewer EKzP Q2), *performance gain* (Reviewer Tdsi W2, Reviewer EKzP W3), and *knowledge aggregation* (Reviewer Tdsi Q1). The paper has been revised accordingly.

We hope our clarifications fully address the concerns, and we appreciate your time.

---

### Meta-Review · Area_Chair_8PNv · 2026-01-07

**Summary:**

The paper proposes knowledge purification to address knowledge conflicts in distillation. The paper received mixed scores. Reviewers raise some concerns, and the authors address part of the concerns in the rebuttal.

**Reviewer Concerns:**

Concerns Addressed or Partially Mitigated by the Rebuttal:
- Scalability to More Teachers (Bt3j, Tdsi):
- Task Generality / Beyond QA Tasks (Wb1Y, Bt3j):
- Lack of Quantitative Efficiency Evidence (Wb1Y):
- Incomplete Baseline Comparisons (Wb1Y):
- Exclusion of TwT Baseline (Tdsi):
- Performance of Knowledge Aggregation (Tdsi):

Concerns Still Outstanding or Not Fully Resolved:
- Limited Theoretical Innovation / Incremental Nature (EKzP, implied by others):
- Fundamental Distinction from Ensemble Methods (EKzP):
- Full Scalability and Generalization (Bt3j, Wb1Y - Residual Concern):

**Reviewer Scores:**

- Reviewer EKzP (Original Score: 4)
- Reviewer Bt3j (Original Score: 6)
- Reviewer Wb1Y (Original Score: 4)
- Reviewer Tdsi (Original Score: 6)

---

### Decision · Program_Chairs · 2026-01-26

Accept (Poster)